# 3D-printed cellular tips for tuning fork atomic force microscopy in shear mode

Liangdong Sun [1,2], Hongcheng Gu [1,2], Xiaojiang Liu[1,2], Haibin Ni [1,2], Qiwei Li[1,2], Yi Zeng[1,2], Ning Chang[1,2], Di Zhang[1,2], Hongyuan Chen [3], Zhiyong Li[1,2], Xiangwei Zhao [1,2✉] & Zhongze Gu [1,2✉]

Conventional atomic force microscopy (AFM) tips have remained largely unchanged in nanomachining processes, constituent materials, and microstructural constructions for decades, which limits the measurement performance based on force-sensing feedbacks. In order to save the scanning images from distortions due to excessive mechanical interactions in the intermittent shear-mode contact between scanning tips and sample, we propose the application of controlled microstructural architectured material to construct AFM tips by exploiting material-related energy-absorbing behavior in response to the tip–sample impact, leading to visual promotions of imaging quality. Evidenced by numerical analysis of compressive responses and practical scanning tests on various samples, the essential scanning functionality and the unique contribution of the cellular buffer layer to imaging optimization are strongly proved. This approach opens new avenues towards the specific applications of cellular solids in the energy-absorption field and sheds light on novel AFM studies based on 3D-printed tips possessing exotic properties.

[1] State Key Laboratory of Bioelectronics, School of Biological Science and Medical Engineering, Southeast University, 210096 Nanjing, China. [2] National Demonstration Center for Experimental Biomedical Engineering Education, Southeast University, 210096 Nanjing, China. [3] State Key Laboratory of Analytical Chemistry for Life Science, School of Chemistry and Chemical Engineering, Nanjing University, 210093 Nanjing, China. ✉email: xwzhao@seu.edu.cn; gu@seu.edu.cn

The invention of tuning-fork-based atomic force microscopy (AFM) has dramatically contributed to the nanoscale visualization of sample landscapes[1,2]. The tips are brought into frequent intermittent contact with the surface interpreting interfacial details from its motion state. However, in this case, the measurement reveals the topography of an artificial surface interacted with the tip rather than its true topography[3]. As a solution, in conventional AFM techniques, the forces applied to the surface from the tip can be accurately controlled through optical lever feedback[1,4] or purposely engineered tip stiffness[5] by tuning the cantilever properties, such as the constituent materials[6,7] or the physical geometries[8]. However, in shear-force feedback, optical lever configuration and cantilever tuning are not applicable since the cantilever is perpendicularly located upon the sample surface with tips vibrating parallel to the sample surface. Geometrical modification of the cantilever is not an effective way to optimize the collision in the orthogonal direction that induces surface deformation and image degradation[9–13], especially at a high setpoint value. As an alternative method, decreasing the setpoint can yield a decrease of vibration amplitude and thus reduces the interactions, which, however, requires a compromise with image quality (e.g., imaging precision, spatial resolution, and signal-to-noise ratio) since the decrease of the setpoint dramatically increase the system sensitivity to mechanical perturbation. In addition, the setpoint cannot be dynamically adjusted during the scanning of complex landscapes featuring steep-slope patterns where feedback "miss" of the setpoint has been reported with continuous undershoot and overshoot[11]. Therefore, it is desirable to complement AFM system with new material constructing piezoelectric tuning fork scanning tips for the alleviation of challenging mechanical interactions between the tip and the sample.

As a succession to natural instances in motion deceleration, shockwave suppression, and mechanical force reduction[14], porous structures widely found in biological skeletal systems such as cancellous bones have been extensively investigated in numerous energy-absorbing applications[15–18]. Emulating these geometrical constructions and coupling with advanced additive manufacturing techniques in microscale, artificial cellular microarchitectures, referred to as controlled microstructural architectured (CMA) material[19–21], can be structurally programmed with a controllable geometry and spatial configuration for advantageous size-dependent metamechanical properties[22,23], such as low density but strong robustness[24], high stiffness-to-weight ratio[25], excellent resilience[26,27], mechanical tunability[28,29], and in particular, energy absorption[30–33]. Hence, by employing this cellular hierarchy for the geometric design of the tip itself, the tip–sample interaction is anticipated to be reduced.

In this paper, we propose the application of CMA material in AFM tip construction based on shear-force-imaging mode to reduce the mechanical impact that the sample surface is subjected to from scanning tips during the scanning process and hence improves the overall imaging quality. The CMA body serves as a compressive media to passively mitigate impulsive loads exerted on the sample surface from an approaching tip by storing and dissipating part of its kinetic energy through the buckling of lattices[34], which reduces the average contact forces and prevents the surface from significant physical distortions by the mechanical impulse[35,36]. The design of the CMA tips follows the guideline of stiffness-matching rule between the tip and the sample. State-of-the-art three-dimensional (3D) direct laser writing (DLW) is employed in tip fabrication[8,32,37] for tailored stiffness[38] due to its arbitrary structural programmability and high precision in three dimensions[37,39–42]. The scans by CMA tips systematically demonstrate positive contributions in optimizing the overall imaging quality. After a further decrease of the stiffness by shrinking the cross-sectional size of struts through post-manufacturing of reactive ion etching (RIE), the CMA tip was applied in the imaging of biological cells with reduced scratches during repetitive scans. The unique concept of the CMA-constructed AFM tips and its superior performance offers unprecedented opportunities in versatile AFM applications.

## Results

**Design and fabrication of the CMA cellular body**. A diamond lattice construction is taken as a reference for the geometrical configuration of the CMA body (Fig. 1a). The unit cell is created in the form of a regular tetrahedron with four vertexes connected to its center while removing the connectivity of its peripheral boundaries. With nanolattices hierarchically stacking over each other, all these units constitute a monolithic CMA object. For controllable mechanical properties, the construction can be flexibly engineered by finely tuning the geometrical variables (e.g., the side length of the unit cell: $a$, referred to as unit side length in the context, as shown in Fig. 1a) in DLW programming. The diamond-structured CMA body was then carefully direct laser written on a fiber facet, which was subsequently bonded to the tuning fork prior to imaging (Fig. 1b, c). The mechanical properties of the assembled probes are provided in the Supplementary Note 1 and Supplementary Table 1. During scanning, the probe tip is advanced and retracted from the sample surface, and moved from point-to-point to map the whole topography of the sample with the apex of the top cell frequently tapping its surface (Fig. 1b). The fabricated shapes of CMA structures by DLW are presented in Fig. 1d–f and Supplementary Fig. 1. The basic voxel generated by one exposure of laser has an ellipsoid shape with lengths of 500 nm and 150 nm for the semimajor and semiminor axis, respectively. Note that the writing paths connecting three feet of the unit cell are obliquely oriented with respect to the major axis of voxels, while the voxels for the top rod is vertically arranged (i.e., parallel to the major axis of voxels), leading to a physical layout of slightly bulky feet sustaining a relatively thin head in each cell (Fig. 1e). As observed from the details in Fig. 1d, e, the CMA body can be perfectly fabricated on the fiber end with an intact shape. The tip in Fig. 1f features a minimum apex radius of ~47 nm, capable of resolving spatial features at a relatively high resolution.

**Mechanical behavior of the CMA structures**. During probe scanning, the mechanical interactions exert a repulsive force between the tip and sample surface, causing the deformation of both. The truss-like foam structure of the tip acts as a compressible buffer to absorb impulsive energy through the bending of the struts[43], which reduces sample deformation during frequent tapping and thereby achieving better imaging precision and contrast. Adverse incidents such as sudden accelerations and shocks caused by environmental perturbation, setpoint "miss", or occasional feedback dysfunction can also be controlled effectively.

Since the energy dissipation mechanism generally comes from the energy-absorbing behavior of the microarchitectures, the compressive behavior of the cellular material was first explored. The compressive response typically includes three stages. In the first elastic regime, the stress increases linearly up to its maximum and then suddenly drops due to plastic yielding in the second stage. The stress remains stable with a plateau curve until densification in the last stage when stress rapidly increases. The energy-absorbing performance of the CMA structure can be evaluated by the peak stress $\sigma_{pe}$, the plateau stress $\sigma_{pl}$, the absorbed energy $w_{absorbed}$ and the efficiency $\eta$, as marked in

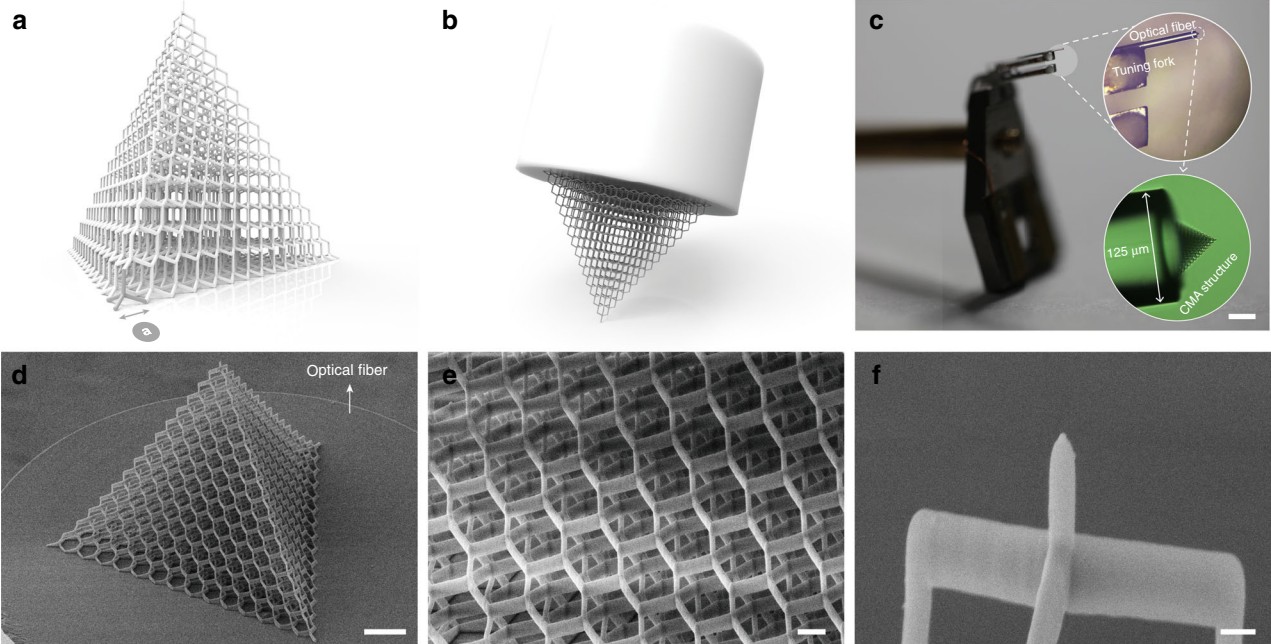

**Fig. 1 Design and fabrication of shear-mode AFM probes based on cellular CMA tips. a** The geometrical design of the CMA body. The mechanical behavior can be tuned by adjusting the unit side length (indicated by the scalar variable "*a*" in the figure and context). **b** Schematic of the CMA structure-based AFM tip. **c** Photographic image of the assembled AFM probe and the microscopic view of its components, which includes a commercial tip mount with a vertical tuning fork, a single-mode fiber, and a cellular CMA structure direct laser written on top of the fiber facet, as shown in the insets. **d**, **e** Scanning electron microscopy (SEM) images of the CMA structures (*a* = 5 μm, 15 stacking layers of cellular units). **f** SEM image of the tip apex with ~47 nm radius. Scale bars are 2 mm, 10 μm, 2 μm, 500 nm for (**c–f**), respectively.

Fig. 2a, which are given as a function of variable strain $\varepsilon$ by:

$$\sigma_{\text{plateau}} = \frac{\int_{\varepsilon_{\text{pe}}}^{\varepsilon} \sigma(\varepsilon)\mathrm{d}\varepsilon}{\varepsilon - \varepsilon_{\text{pe}}} \quad \varepsilon \in \left[\varepsilon_{\text{pe}}, \varepsilon_{\text{tr}}\right] \quad (1)$$

$$w_{\text{absorbed}} = \int_{0}^{\varepsilon} \sigma(\varepsilon)\mathrm{d}\varepsilon \in [0, \varepsilon_{\text{tr}}] \quad (2)$$

$$\eta = \frac{w_{\text{absorbed}}}{w_{\text{ideal}}} = \frac{\int_{0}^{\varepsilon} \sigma(\varepsilon)\mathrm{d}\varepsilon}{\sigma_{\text{pe}}\varepsilon} \quad \varepsilon \in [0, \varepsilon_{\text{tr}}] \quad (3)$$

The plateau stress is an approximation of average stress in the second buckling stage ranging from the peak force strain $\varepsilon_{\text{pe}}$ to a given strain value in this regime. The absorbed energy is denoted by an integral area below the stress–strain curve at the given strain range (the hatched area in Fig. 2a). The ideal situation of absorbed energy is defined as the integration of the constant peak stress for a given strain range (the gray area in Fig. 2a), and the efficiency is the ratio of absorbed energy to the ideal energy. Additionally, a threshold of strain is defined as $\varepsilon_{\text{tr}}$ as shown in Fig. 2a and specified with $\varepsilon_{\text{tr}} = 1.8\%$ to calculate the specific values of plateau stress and absorption efficiency within this strain range. The mechanical behavior of CMA structures with different lattice sizes is explored by instrumented indentation tests (IIT). The indentation images obtained from in situ measurements manifest exceptional recoverability and supreme resilience of this engineered material (Supplementary Fig. 2)[26].

Representative measurements of the engineering stress–strain responses of CMA structures with different unit side lengths are shown in Fig. 2b. The engineering stresses are calculated according to the definition and depicted in Fig. 2c. The average plateau stresses of given CMA structures are generally distributed within 0.1–0.54 MPa while average peak stresses are within 0.23–0.72 MPa. The overall energy-absorbing efficiencies are

distributed in the same range of 0.55–0.69 as the values in reported literature[43], showing no significant differences for all tested CMA structures albeit a distinct decreasing trend of the stress indexes (i.e., peak stress and plateau stress) as cell size increases. This feature indicates that the selection of CMA tips should be according to specific requirements of the stress level instead of efficiency. As shown in Fig. 2d, the energy absorption diagram directly relates the absorbed energy to the stress at a corresponding strain value with only the energy amount and the effective stress range concerned. Since the best tip is theoretically defined as the one that absorbs the most energy, the softest tip (*a* = 5 μm) outperforms others in the lowest stress range, but the stiffest tip (*a* = 2 μm) exhibits optimal performance in the highest stress range. Indicated by the colored backgrounds in Fig. 2d, the trend in tip selection suggests that the tip should be designed stiffer with a smaller lattice size as the working stress level increases in order to absorb the most impact energy. Since the allowable stress level transmitted from the tip to the surface in practical applications also increases as the sample itself becomes stiffer, stiffer samples clearly require the selection of a stiffer tip. However, the exact stress range that induces acceptable deformation of the sample interface remains unknown. It is difficult to quantitatively build an accurate correlation of specific stiffness between the sample and required CMA structure solely from the diagram.

Therefore, a dynamic impact simulation of a CMA tip (*a* = 5 μm) and a solid cone tip is proposed for further exploration, as shown in Fig. 2g, h. The tips are approaching at an initial speed of 20 μm s⁻¹ towards the soft substrate and impact the surface. Figure 2h shows the temporal evolutions (0–10 ms) of substrate cross-section in the near-field of contact position, indicating non-uniform interfacial deformation in response to impulsive local stresses occurring during the dynamic impact[44]. For the CMA tip, the indentation depth increases rapidly in the first 6 ms but remains almost constant for the remainder of the time,

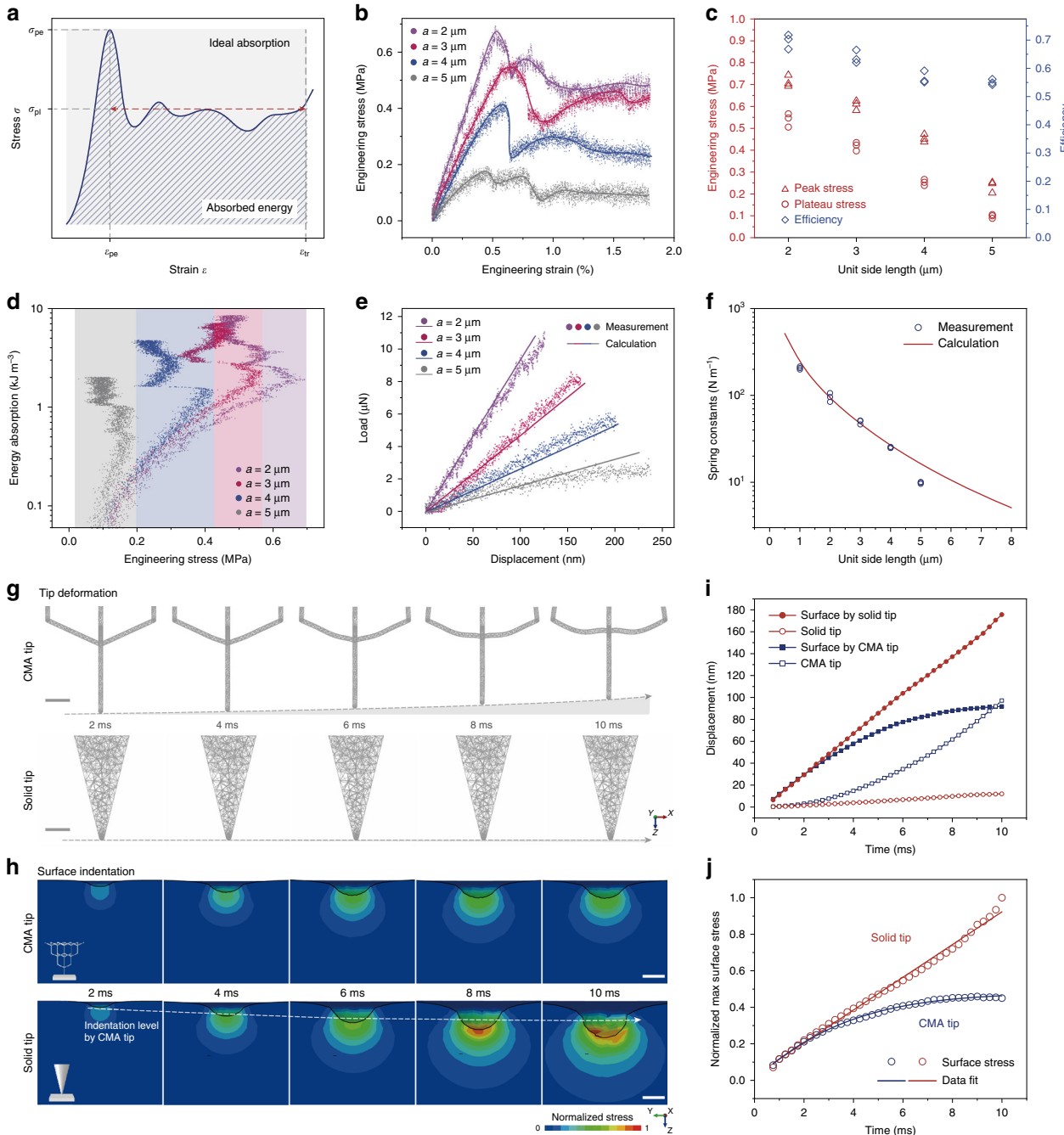

**Fig. 2 Mechanical characterization of CMA structures. a–d** Characterization of the energy-absorbing performance of CMA structures by instrumented indentation tests (IIT). **a** Schematic of typical stress–strain ($\sigma - \varepsilon$) curve of materials in response to compression, with its key features of peak stress $\sigma_{pe}$, plateau stress $\sigma_{pl}$, peak force strain $\varepsilon_{pe}$ and strain threshold $\varepsilon_{tr}$ marked in the plot. **b** Engineering stress–strain features of CMA structures with different unit side lengths obtained from IIT at a strain rate of $10^{-1}\,s^{-1}$. **c** Calculated engineering stress (plateau stress and peak stress) and energy-absorbing efficiency of CMA structures (represented by unit side length). **d** Energy absorption diagram showing the amount of absorbed energy per unit volume as a function of stresses at the given strains. **e, f** FEA simulation of the elastic compressive response of CMA structures based on static loading. **e** Load-displacement features of CMA structures in the linear elastic regime. The solid lines are simulation predictions. **f** Calculated and measured spring constants versus structures with different unit side lengths. **g–j** Dynamic tip–sample impact simulation based on dynamic finite element analysis (FEA). The tips have an initial speed of $20\,\mu m\,s^{-1}$ in $+z$ direction approaching and impacting the sample surface (see Supplementary Movie 1). **g** Morphological evolutions of a CMA tip ($a = 5\,\mu m$) and a solid cone tip during the first 10 ms of the impact. All tips are displayed in a render style of wireframes. **h** Time-lapse indentation cross-sections and normalized local stress distributions on the sample within the first 10 ms (tips are concealed) by employing the CMA tip and the solid tip as shown in the insets. For comparison, the dashed arrow line indicates the evolution of the indentation level by the CMA tip. **i** Tip displacement and max interfacial indentation depth over time. **j** Normalized maximum interfacial stress versus time. Scale bars are $1\,\mu m$ for **g** and 100 nm for **h**. Source data of **b–f**, **i**, **j** are provided as a Source Data file.

whereas the results of a solid probe show deeper indentations growing at a constant rate for the whole period with apparently higher overall stress distribution. As reflected by the attendant morphology of the CMA tip along the timespan in Fig. 2g, the stress that compresses the substrate also leads to evident buckling of the tip, particularly noticeable after 6 ms in comparison to the initial shape. Evidenced by the quantitive description of corresponding displacements of the tip and substrate in Fig. 2i, the compressive media acts as a sacrificial layer to sustain part of the displacement from the indentation path in order to save the substrate from severe surface distortions that would have occurred when a solid tip is used, as shown in Supplementary Fig. 3. In the later period, the interaction between the CMA tip and the substrate reaches a balance. Further indentation is counterbalanced and grows slowly due to a negative feedback loop, as shown in Fig. 2i, since the stress will surely increase if the tip continues pressing into the substrate, causing more "retraction" of the tip due to its compressive nature. The solid tip acts as a rigid body with almost neglected self-deformation during the impulsion (Fig. 2g) and causes the linear growth of indentation depth and stress (Fig. 2i, j). Employing the CMA tip, the maximum local stress on the interface can be controlled under 0.45 after normalized with the maximum results from the solid tip, as depicted in Fig. 2j. This result further translates into a specific requirement of stiffness-matching principle[5] in the design of the CMA structures after a combined consideration of minimizing impact indentation while still keeping the tip deformation in a reasonable range needed to maintain its stability in practical scanning applications. Specifically, if the tip is designed to be much stiffer than the sample, the buffer effect will be restricted, however, if the tip is designed to be much softer than the sample, the imaging will be degraded as well due to the mechanical instability of the scanning tip even though the surface itself is not excessively deformed.

To predict the structure stiffness, static compressive analysis was performed with the apex subjected to uniaxial compression for specific displacements corresponding to the applied stresses (Supplementary Note 2 and Supplementary Fig. 4). The design of a tip with variable stiffnesses comparable to the stiffness of the sample relies on the proportion of voids and solids in the diamond-structured CMA constitution, which is achieved by structural variation of the unit side length and stacking layers of cellular units. However, the latter does not greatly contribute to the overall stiffness performance, especially when exceeding a particular amount (Supplementary Fig. 4c). For practical engineering, solely adjusting the unit side length for tunable mechanics is preferred. Therefore, all the CMA tips applied and discussed in the following AFM imaging tests were fabricated with default 15 layers. Figure 2e gives the load-displacement features of CMA structures with different lattice sizes, with the solid lines indicating the simulation predictions. The estimated spring constants are calculated and plotted in Fig. 2f. The experimental results are in good agreement with the simulated values, suggesting the numerical predictions can be potentially utilized as a reference to tailor specific tip design with a designated stiffness. According to the calculated values, the spring constant of the structures can systematically cover a range from 4.2 to 386 N m⁻¹, over two orders of magnitude.

Importantly, this CMA design of diamond lattice constructions in this study is merely one example of myriad complex hierarchical cellular structures. The tuning range of stiffness are unlimited and can be readily varied by the modification of engineered composites[45,46], geometries as well as shapes[26] according to specific applications, opening paths towards unrestricted practical applications.

**AFM imaging on silicon microgrids with CMA tips**. To validate the essential imaging functionality using the cellular tips and to verify whether the buckling of the tip itself, which is intended to absorb impulsive energy for the buffer effect, would pose a negative influence on the accurate height measurement, microgrids made of silicon material (Young's modulus: ~190 GPa) with a nominal step height of ~113 nm were first used for calibration. Regarded as a rigid object due to its overwhelming stiffness, interfacial deformation of the sample can be neglected so that any difference in measurement results only comes from the specific situations of corresponding tips. Two CMA tips with different unit side lengths ($a = 1\,\mu m$ and $a = 5\,\mu m$) and a commercial tip with a nominal spring constant of 2600 N m⁻¹ were applied. With larger lattices ($a = 5\,\mu m$), the softer tip stiffness is approximately four orders of magnitude less than that of the substrate. In contrast, the stiffer tip ($a = 1\,\mu m$), possessing a relative density of 0.85 due to closely packed lattices, is better considered as a solid containing pores rather than a true cellular solid[47]. This tip behaves more like a pure solid counterpart and is thus well-suitable for sampling rigid surfaces[8,37] that do not have a pressing need for a mechanical buffer during scanning.

Figure 3a provides the height contours imaged by three tips in the same region of the microgrids. The basic scanning capability by the cellular tips can be preliminarily demonstrated by mutual consistency in scanning figures. To further quantify the accuracy of measurements, the height profiles along the same four white paths marked 1–4 (Fig. 3a) is shown in Supplementary Fig. 5. According to the fitted height statistics of imaged step patterns in Fig. 3d, the data obtained by the CMA tips coincides perfectly with the calibrating record from the commercial tip, and the height error was kept within 4 nm, a relative deviation within 3%. Theoretically, as a relative characteristic, the height is the morphology difference between the upper stage and the lower ground. Considering a stable stress output at overall locations in a single mapping after the scanning process is stabilized, the tip deformation or approaching distance of a given tip is theoretically constant at both positions. The height mapping produced by the subtraction of approaching distances would be the same, independent of the specific lattice sizes and possible deformation of the tip itself as revealed by the measurement consistency, suggesting the reliability of CMA tips for precise height measurement.

However, the softer tip ($a = 5\,\mu m$) bends much more than the stiffer tip when hitting a rigid surface during scanning, which renders an unstable movement in the approaching and retracting process and, therefore, featuring a relatively lower contrast in the height plot with significant artifacts occurring especially when climbing the edges (indicated by dashed circles in Fig. 3a). The noticeably blurred edge profiles accompanied by the plenty of noise signal lines (pointed by black arrows in Fig. 3a) that are derived from contact feedback loss suggest the probe's susceptibility to environmental perturbation[12], as compared to the imaging results by stiffer tip ($a = 1\,\mu m$). Specifically, the motion details of the oscillating tuning fork are revealed in the mapping of phase and amplitude (Fig. 3b, c). As reflected by the obscure profile with double fringes in the phase plot (indicated by black arrows in Fig. 3b) as well as double amplitude peaks (inset data curve in Fig. 3c) along step pattern (white arrow lines in Fig. 3c) in the amplitude plot (Fig. 3c), the unstable motion state of the softer tip ($a = 5\,\mu m$) in the scanning course can be strongly proved, cueing a possible transition of the tip movement at the descent trace.

Therefore, the height variation along the edges (white arrow lines in Fig. 3a) was extracted and investigated. As sketched in Fig. 3e, the tip motion is characterized by two phases due to the

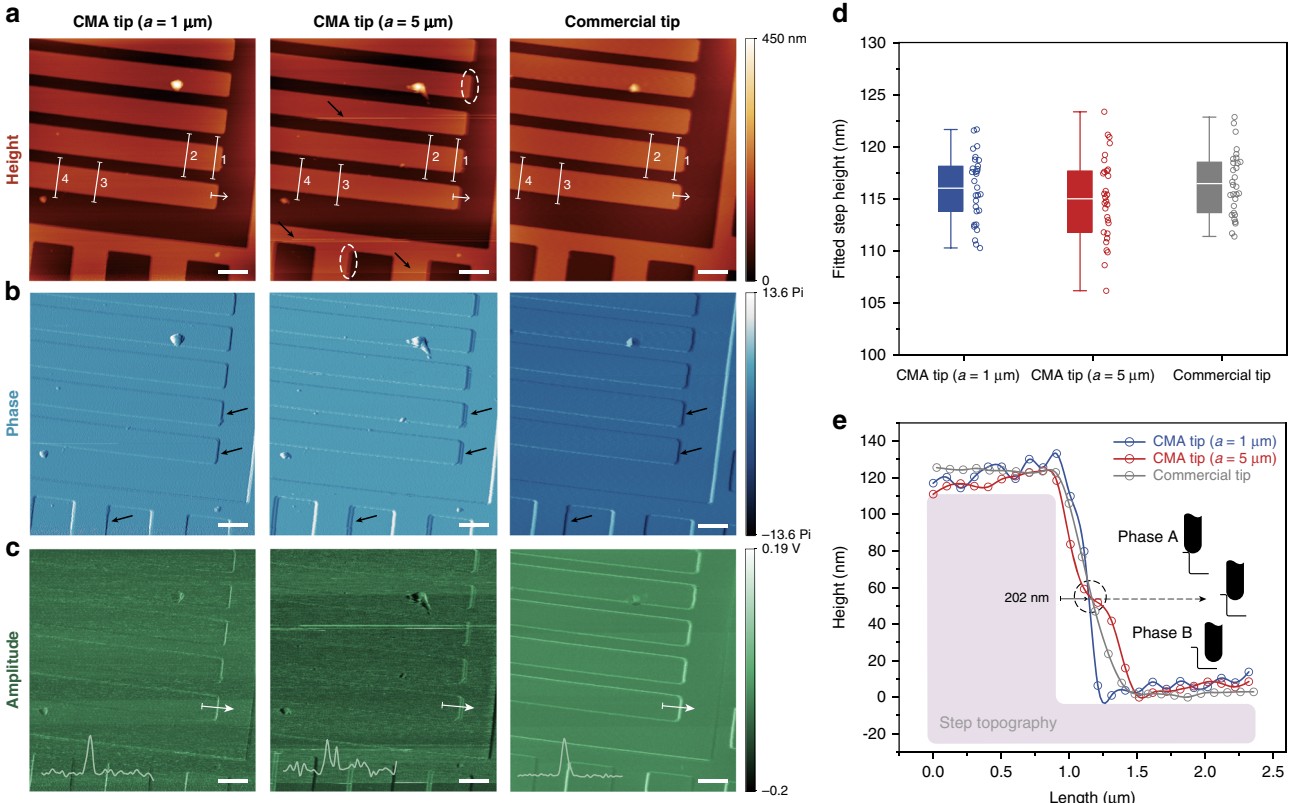

**Fig. 3 AFM imaging of silicon microgrids with CMA tips. a–c** AFM maps of height, phase and tuning fork amplitude by employing CMA tips ($a = 1\,\mu m$ and $a = 5\,\mu m$) and a commercial tip, respectively. The black arrows in **a** indicate the signal noise derived from unstable movement, while the black arrows in **b** show the imaging difference of the grid edges by different tips. The inset signal curves in **c** denote the amplitude voltage of the tuning fork when crossing the step edges (white arrowed lines). **d** Boxplot of step height acquired by the CMA tips and the commercial tip. The Boxplot marks the median (center line within box), the first and third quartile (box), and 1.5 times the interquartile range (whiskers). The corresponding data are exhibited as scatters on the right side of each box. **e** Height profile of the step pattern along the white arrow lines marked in **a**. The scanning processes of the tips are divided into two phases (Phase A and B) as sketched by the insets with transition point indicated by the dashed circle. All scale bars are 4 μm for **a–c**. Source data of **d**, **e** are provided as a Source Data file.

perfect similarity in Phase A whereas a distinct deviation in Phase B in terms of measured height curves. The transition point (indicated by the dashed circle) measured to be 202 nm away from the cliff is approximately the cross-sectional radius of the top rods of CMA structures. The trail of the soft tip undergoing a slight bounce from the cliff right at this position likely arises as a consequence of tangential contact between the tip side and the step wall, which can account for the double states when crossing the edges as recorded in phase/amplitude maps and serve as another evidence of probe instability originating from the mismatched stiffness of the tip and the sample. It might not influence the accuracy of step height measurements but would decrease the image contrast of critical features and performance of signal-to-noise ratio. Therefore, considering the buffer effect and practical experience from the above experiments, the tip design should be customized with a similar stiffness range to the sample. Additional scanning images of the silicon microgrid are provided in Supplementary Fig. 6.

**AFM imaging on polydimethylsiloxane (PDMS) patterns with the CMA tips.** To demonstrate the desired buffer functionality from the CMA construction for optimized quality in imaging a soft substrate, micropatterns made of PDMS were prepared for tests. The crosslinking ratio of PDMS agents is 10:1, which leads to a spring constant of 0.3–5 N m$^{-1}$ or a material modulus of 2.6 MPa[48–52]. The microfeatures were molded from a two-dimensional (2D) DLW helical pattern using a template

stripping method (Supplementary Fig. 7a–c)[53], with a groove width calibrated to be $192 \pm 13$ nm by scanning electron microscopy (SEM), as shown in Fig. 4a. For the scanning configuration, both the CMA tip ($a = 5\,\mu m$) and the non-CMA tip (solid cone shape with a modulus of ~2.5 GPa) were selected for imaging comparison. Here, the same DLW technique and writing parameters were adopted for tip fabrication to ensure similar sizes and shapes of the tip apexes, in order to minimize possible imaging difference induced by apex variation (Supplementary Note 3). The scanning for both tips was maintained at a similar but increased setpoint value (0.35–0.4) compared to the typical requirement in soft sample imaging (Typical setpoint: 0.15–0.25) to achieve a more violent tip approach (Supplementary Note 4).

The images produced by the CMA tip and the solid tip are presented in Fig. 4b. From the 2D height contour, both the features of the tunnels and surface grains obtained by the CMA tip are recognizable with a much flatter surface, while scanning images by the solid tip are severely degraded associating with a significant expansion of the groove features as well as violent interfacial height fluctuations. The obvious overall color change along the slow scanning axis can be visualized in the height plot acquired by the solid tip, indicating steep ascents and descents across the landscape as illustrated by the reconstructions of the corresponding 3D topography. To quantively characterize the imaging difference on the surface, representative regions of interest (color blocks marked with "A" and "B" in Fig. 4b) are extracted for analysis in Fig. 4c, which reveals different homogeneities in height distribution. The

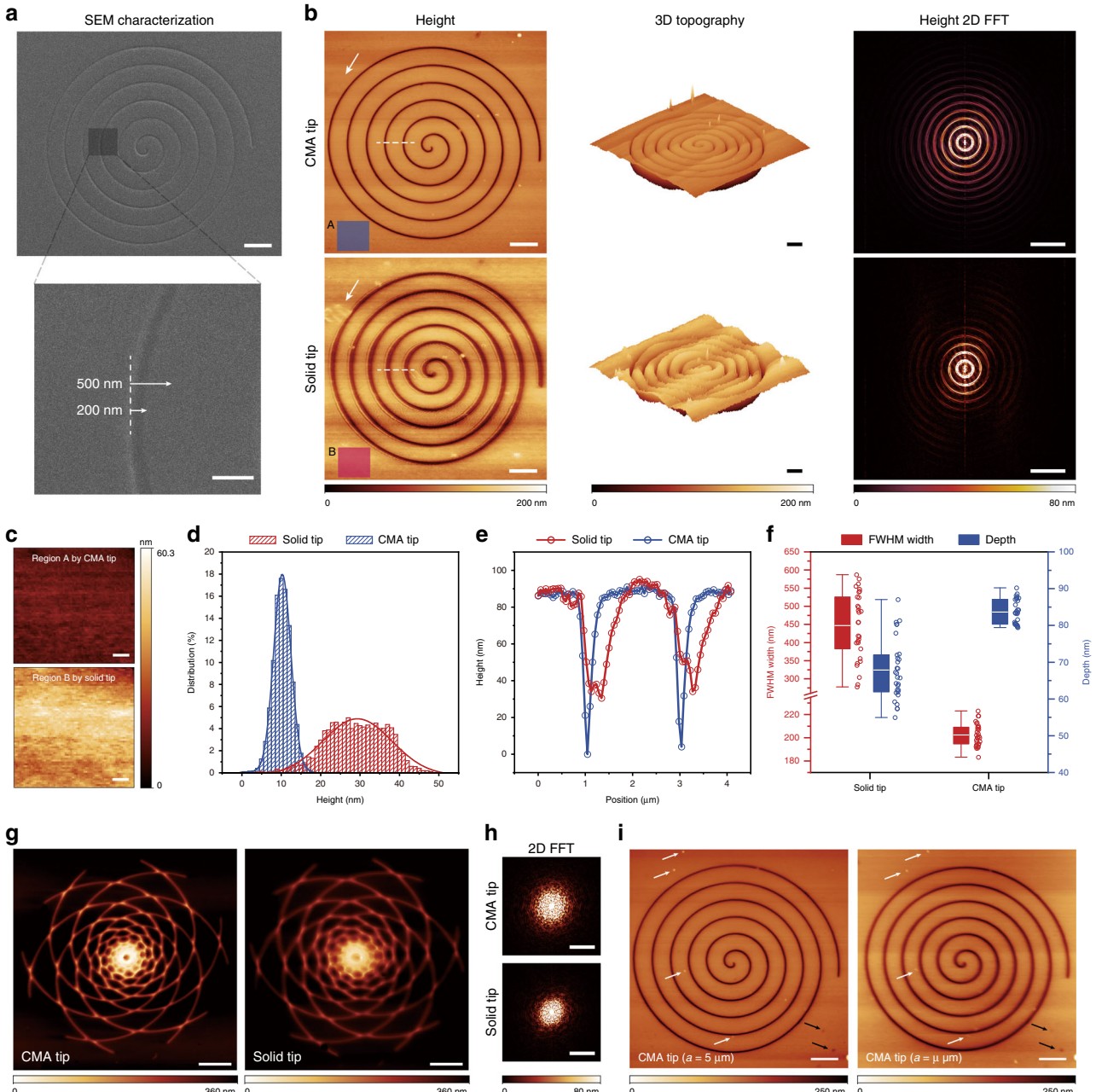

**Fig. 4 AFM imaging of PDMS patterns with CMA tips. a** SEM images of PDMS spiral patterns molded from the DLW 2D template (Supplementary Fig. 7c). The lengths of the white pointed lines in the SEM image represent physical distances of 200 nm and 500 nm, respectively. Scale bars are 3 μm and 500 nm for upper and lower images, respectively. **b** Scanning images respectively acquired by the CMA tip ($a = 5$ μm) and the solid tip (DLW solid cone) with images from left to right respectively corresponding to 2D height plots, 3D topographies, and 2D fast Fourier transformation (2D FFT) images of the height plots using a Hanning window. The same color bars are used for height, 3D topography, 2D FFT images, respectively. Scale bars are 3 μm for height and 3D topography images and 3 μm⁻¹ for 2D FFT images. **c** Magnified surface details (scale bar is 500 nm) separately extracted from region A and B in the height plots of **b**. **d** Histogram analysis of interfacial landscape in **c** showing frequency counts (1 nm step) of height distribution. The histograms are fitted with Gaussian curves. **e** Height profiles along the white dash lines in **b**. **f** Boxplots of FWHM width and depth of the groove measured by the solid tip and CMA tip in **b**. The Boxplot marks the median (center line within box), the first and third quartile (box), and 1.5 times the interquartile range (whiskers). The corresponding data are exhibited as scatter plots on the right side of each box. **g, h** Height profile (**g**) of a flower pattern (Supplementary Fig. 7d, e) and the corresponding 2D FFT images (**h**) obtained by the CMA tip ($a = 5$ μm) and the solid tip (DLW solid cone) at a decreased setpoint value. **i** Imaging comparison using CMA tips with different unit side lengths ($a = 5$ μm and 2 μm). The surface grains are indicated with white and black arrows. Scale bars are 3 μm for **g, i** and 3 μm⁻¹ for **h**. Source data of **d–f** are provided as a Source Data file.

upper image displays a more uniform result while the other one features a discrete distribution with violent fluctuations, especially along the slow scanning axis. According to the height histogram described in Fig. 4d, the CMA tip scanned surface image displays a range of 18 nm in height distribution, possessing an average value of 10.3 nm and an RMS (root mean square) roughness of 2.2 nm. By contrast, the presented surface image by the solid tip manifests a dispersed distribution widespread over a range of 51 nm with an average height value of 29 nm and an RMS roughness of 7.6 nm. Theoretically, in the ideal situation, the exerted stress from the tip

would be constant at each scanning point, leading to the same indentation depth at every point of the surface correspondingly, if the constituent material of the whole sample is homogenous, which should thus have yielded a similar roughness distribution by two tips even though the absolute value of height differs. Since the measured area is identical for both tips and the stress is stable during the scanning process, the only accountable reason for the roughness difference is rooted in the interfacial inhomogeneity of stiffness at the nanoscale. With the solid tip, the stress, maintaining a relatively high level, induces a prominent local strain difference among the interfacial positions that even though possess minor divergences of stiffness. In fact, the measured information by the solid tip highlights the stiffness difference of the sample rather than reflecting the real topography. Specifically, in the height plot of Fig. 4b, the imaging deviation of the area indicated by the white arrow can be readily discerned. The light-colored contours obtained by the solid tip correspond to a stiffer region distinct from the surrounding neighborhood, which rises as a highland after tip interaction. For a CMA tip, the apparent small strain difference for inhomogeneous regions can be interpreted as evidence of a sharp decrease of interactive stress in comparison to the solid tip, which prevents the inhomogeneous surface from severely discordant deformation and hence retains its original surface morphology. In Fig. 4b, the height images are processed by 2D fast Fourier transform (2D FFT) to clarify the distortions, roughnesses, and artifacts[6]. The lower FFT image obtained by the solid tip shows broken symmetry with discontinuous patterns distributed at the outside boundary, while perfect overall symmetry is maintained for the CMA tip. The groove width and depth are also presented as a judgment of image quality. The height variation obtained by the CMA tip along the white dash lines in Fig. 4b features a steeper depth but a smaller width (Fig. 4e). The boxplot in Fig. 4f further presents quantitative characterization for the FWHM (full width at half maximum) width and depth of the grooves. The width is 202.4 ± 9.7 nm statistically analyzed from the scanning results of the CMA tip, which is raised to 446.9 ± 91.2 nm with a significantly increased discretization by the solid tip. The depths are 83.6 ± 3.3 nm and 67.9 ± 8.2 nm given by the CMA tip and solid tip, respectively. The width measured by the CMA tip is consistent with the SEM observation, while the width yielded by the solid tip is substantially broadened with twice as much. It is also noteworthy that there is a deviation in depth measurement, which likely arises as the same result of different stress levels, allowing us to theoretically estimate the corresponding transient stresses applied by tips in the dynamic impact of the deceleration phase. According to the calculation outcomes provided in the Supplementary Note 5, the average impact-derived local stress level applied by the CMA tip is ~0.07 MPa, which is about 9.8%-18.3% of the stress produced by the solid tip.

In Fig. 4g, the setpoint value was then decreased (details in "Methods") to determine whether the adjusting of the feedback control parameters could significantly optimize the imaging artifacts induced by the solid tip. The flower-patterned PDMS mold is composed of rotating golden helix curves with a protruding point located at the center (Supplementary Fig. 7d, e). By employing a solid tip, the center point is so greatly suppressed down toward the low ground that it is challenging to distinguish it from adjacent interfaces. In Fig. 4h, the FFT image by the solid tip is distorted in the direction of the slow scan axis, which underlines the negative distortion and roughness of the imaged surfaces as well as the related excessive stress induced by the solid tip. Quantitatively, the sizes of the smallest distinguishable features are ~60–80 nm by the CMA tip, while the smallest patterns are distributed in the range of ~300–600 nm using the solid tip. Considering that there are 500 × 500 points in a 21 × 21 μm² region, the step length is 42 nm. The spatial resolution of the CMA tip-scanning result is rather close to

the positioning limit based on the current configurations (Supplementary Note 9). Therefore, the precise measurements and significantly improved image contrast (Fig. 4g, h) inaccessible by solely adjusting feedback control parameters conversely demonstrate the effective contributions by the CMA designs. More tests are recorded and analyzed in Supplementary Note 6.

The imaging difference acquired using CMA tips with two different lattice sizes is presented in Fig. 4i to verify the contribution of the stiffness-matching principle in tip design to imaging optimization. Two CMA tips were applied to image the PDMS spiral pattern. The softer tip has a unit side length of 5 μm with an ultralow relative density of ~0.03, featuring an average spring constant is 9.6 N m⁻¹, which is more comparable to PDMS stiffness. For the stiffer tip ($a = 2$ μm), the relative density is ~0.2 due to the shrinkage of interior void space, which is about the maximum limit of typical polymeric foams used for cushioning and insulation (Supplementary Fig. 12)[47]. The spring constant is correspondingly increased to 84.2 N m⁻¹, one order of magnitude stiffer than the substrate. Compared with the solid tip-scanning results shown in Fig. 4b, the image obtained by the stiffer CMA tip ($a = 2$ μm) is still blurred on the left and right sides, albeit a visual optimization in surface roughness and groove measurement. Tiny features of surface grains indicated by the white and black arrows are not very clearly outlined compared with the corresponding locations in the softer tip-scanning image, suggesting that the exerted stress is still maintained at a relatively high level. The results further underpin the necessity of stiffness-matching principle in engineering practical CMA designs. Multiple experiments have been conducted to validate the repeatability and reliability of imaging performance by the CMA tips, as shown in Supplementary Fig. 13.

In addition, some experiments (Supplementary Fig. 11) also suggested that the stress for each point was not perfectly stable over the course of the experiment, and unreal grains were produced due to occasional overstressed situations. In practical operation, the drift of the error signals and the feedback deviation from the initial setup is almost inevitable due to long-term scanning (e.g., a typical 400 × 400 points scanning with 7 ms per point requires over 30 min), which requires a real-time manual calibration. With CMA material as a buffer layer, the stress perturbation can be stabilized within a small indentation of the sample, which allows for the fluctuation of the error signal over a relatively wide range. Images obtained by CMA tips imitating situations at increasing setpoint values are provided in Supplementary Fig. 14. Stable scans can be thus guaranteed over this long period avoiding the fluctuation of the error signal or stress level, thereby eliminating the need for further realignment and achieving stable and accurate images. In the entire study, the scanning with CMA tips was conducted over one hundred times, and not a single case of tip fracture was recorded (Supplementary Note 7). More scanning images of soft samples by the CMA tips are provided in Supplementary Fig. 16. Further exploration of normal-mode imaging with CMA tips has also demonstrated success, as revealed in Supplementary Note 8. The application of the 3D-printed CMA tips is far more feasible and durable than we had expected, which is perfectly suitable for intermittent contact imaging (Supplementary Note 9).

**AFM imaging of cells with the CMA tip**. Multiple post-manufacturing methods have been applied to modify the mechanical behavior of CMA structures (Supplementary Note 10 and Supplementary Figs. 18–20). In order to further reduce the tip stiffness to match the stiffness of biosamples, RIE[54] was employed to conformally shrink the beam size of a CMA tip ($a = 5$ μm), which is accurately controlled as a function of etching duration, as shown in Supplementary Fig. 20. After processing for

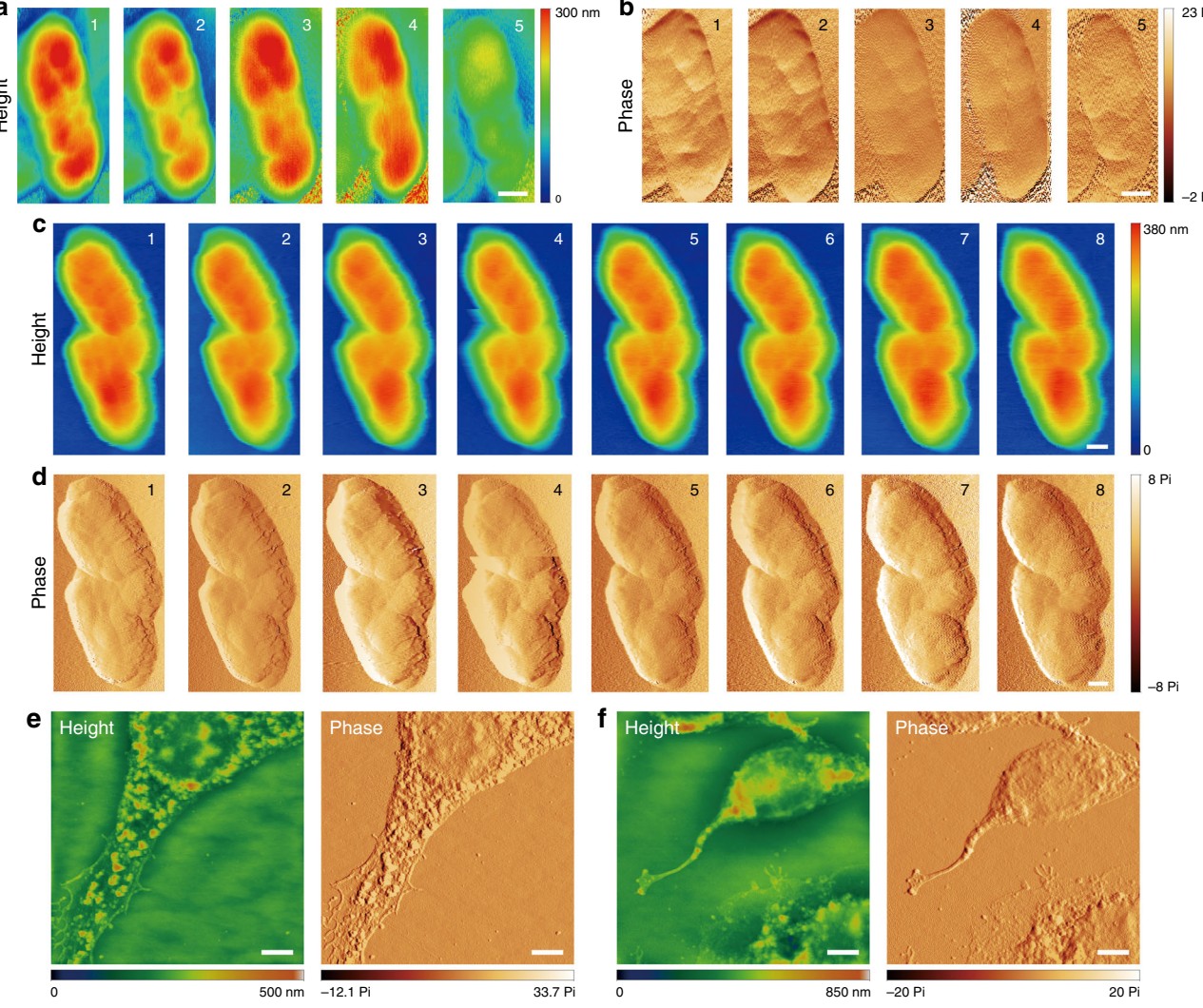

**Fig. 5 AFM imaging of biosamples with RIE-etched CMA tip ($a = 5\,\mu m$). a**, **b** Height and phase profiles of *Shewanella oneidensis* MR-1 bacterial cells subjected to repetitive scans by a commercial tip. **c**, **d** Height and phase profiles of *Shewanella oneidensis* MR-1 bacterial cells after repetitive scans of an RIE-etched CMA tip ($a = 5\,\mu m$) (Supplementary Fig. 20). The scanning times are marked at the top right corner of each figure. **e**, **f** AFM imaging of fibroblasts with the etched CMA tip ($a = 5\,\mu m$). Scale bars are 400 nm for **a**–**d** and 5 μm for **e**, **f**.

60 s, the structure still retains perfect shape achieving an average vertical beam diameter of 105 nm and an apex radius of average 23 nm to a minimum of 17 nm (Supplementary Fig. 20). Further extending the RIE duration will surely help in shrinking the size but endure a high risk of structural fracture. The spring constant of the tip was estimated to be ~0.12 N m$^{-1}$ after all measured size parameters were introduced into the calculation model. The *Shewanella oneidensis* MR-1 bacterial cells (spring constants: 0.02–0.05 N m$^{-1}$)[55,56] were selected for imaging with this etched CMA tip, and successful results revealing the height and phase landscape of the specimen are provided in Supplementary Fig. 21, which depicts the unambiguous contours of the microbe.

To further explore the performance of superior mechanical cushion effect enabled by CMA tip, repetitive scan tests[6] under enhanced interactions induced by purposely increased feedback control parameters were proposed with a commercial probe taken as a comparison. Height images painted with rainbow false-colors and corresponding phase information are presented in Fig. 5a–d to determine the slight changes in surface topography among the scanning times (scanning time is numbered on the top right corner of each picture). Employing a commercial tip, the nanofeatures of the cell gradually disappear with apparent wear

patterns and distortions after repetitive scans (Fig. 5a, b). Obvious scratches on its surface can be visualized since the 2$^{nd}$ scan, and the average height of the cell surface decreases by ~100 nm after the 5$^{th}$ scan compared with the initial measurement, arising from overwhelming stress imposed by the tip. In contrast, the overall features of the cell were well-retained even after the 8$^{th}$ scan of the etched CMA tip with height and phase maps remaining relatively consistent throughout repetitive scans (Fig. 5c, d). Notably, the surface scratches are not obvious until the 6$^{th}$ scan. Building upon these results, the application of the CMA tip conclusively exhibits better performance for biosample imaging than commercial products under current configurations. More AFM images of fibroblasts (spring constants: 0.2–0.4 N m$^{-1}$)[57] by the CMA tip can be found in Fig. 5e, f. The distinct details in height and phase plots clarify not only the cell morphology but also the extraordinary performance of the CMA tip, which will surely be of benefit to a wide range of engineering fields.

## Discussion

In summary, we have proposed a special tip design for the AFM probe based on hierarchically constituted microstructural

architectures by utilizing the structure-correlated mechanics. The tip serves as an impact-resistant component to alleviate the impulse intensity from the tip to sample that degrades the imaging quality during scanning. Specifically, suggested by finite element analysis of structural response, the stiffness-matching principle between tip and sample was established for the design of requisite structures, which was readily addressable via DLW. By solely adjusting the cell size, tunable structural spring constants over two orders of magnitude were achieved. The viability of the concept is systematically proved through multiple scanning of samples, including stiff silicon grid, soft polymer patterns, and biological cells, with pronounced optimization in imaging quality. The innovative integration of CMA buffer material with the tuning fork AFM system opens up an alternative path for tip-controlled imaging scheme and is expected to beneficially facilitate versatile scanning probe microscopy studies based on 3D-printed tips for both industrial and academic use.

## Methods

**Fabrication of CMA tips**. Direct laser writing was employed for the manufacturing of the CMA structures on the fiber facet, which is based on a commercial GT system (Photonic Professional GT, Nanoscribe GmbH). The femtosecond laser is centered at 780 nm wavelength with a pulse duration of 100 fs. The negative photoresist for two-photon polymerization was IP-Dip (Nanoscribe GmbH). The data script of accurate voxel position used for writing was coded in MATLAB® software, and piezo scan mode was applied for precise printing. The settling time for stage and piezo was maintained at 300 ms. The manufacturing process for a single CMA structure ($a = 5$ μm) with default 15 layers of units lasts about 25 min at an automatically adjusted writing speed of ~40–60 μm s$^{-1}$ for perfect shape quality. To prepare the fiber platform before printing, single-mode fiber was cut using an optical fiber cleaver followed by 30 min ultrasonic bath in acetone and then dried with nitrogen flow. The fiber was immobilized on a specially customized holder with the flat facet carefully immersed into the photoresist droplet that was previously dropped onto the objective lens (63×, NA1.4, Zeiss). After fabrication, the exposed resist was developed by 30 min PGMEA bath and 10 min n-pentane bath in sequence. The optical fiber with CMA structure located at one top was bond to the tip mount (MV4000N5 TF Tip Mount, Nanonics Imaging Ltd.) with UV-cured adhesive (8500 Metal, ergo) before imaging. The adhesive can be easily removed with 3 min ethanol bath so that the fiber can be readily attached and detached.

**Simulation settings for the mechanical analysis of CMA structures**. Mechanical simulations based on both dynamic and static loading were conducted to explore the compressive response of CMA structures in this study. The dynamic impact process between the tip and the sample surface was simulated in ABAQUS/Explicit based on FEA. A CMA tip ($a = 5$ μm) was employed in the simulation with a solid cone tip (a taper angle of 28°) applied for comparison. Both tip apexes were designed with an equal radius of 100 nm. Both models were built in Solidworks and subsequently synchronized to the simulating software for analysis. The modulus of the tip and sample material were specified as 2.5 GPa and 50 MPa, respectively. The tips were initially placed 10 nm above the surface with a relative speed of 20 μm s$^{-1}$ approaching the sample. The local deformation and max stress change over the first 10 ms are presented in Fig. 2g, h, and Supplementary Movie 1.

The static behavior of the CMA structures was performed in COMSOL Multiphysics using finite element calculation. The model coordinates were generated in MATLAB® and imported into COMSOL, followed by specific cross-sectional assignments of the struts (Supplementary Fig. 4a). The properties of structural material were specified with a Young's modulus of 2.5 GPa and a Poisson's ratio of 0.49. The cross-section for the head is a circular shape of 0.4 μm diameter, while for the feet, it is a rectangle shape of $1.1 \times 0.4$ μm$^2$. Structure gravity was considered. Extra force (0–10 μN) was applied to uniaxially compress the apex, and the stiffness properties were determined from the relations of force and displacement.

**Indentation tests**. Indentation tests were performed on PI 85 SEM PicoIndenter from Hysitron® company. The stiffness data of structures was calculated from the elastic scale of the corresponding load-displacement curves. The engineering stress and strain were calculated from load-displacement curves with cross-sectional footprint area and structure height, respectively.

**Parameter settings, tip selections, and sample fabrications in AFM imaging**. All AFM imaging experiments were performed on a multi-probe scanning probe microscopy system (MV4000, Nanonics Imaging Ltd.) at shear-force-imaging

mode with phase feedback. For stable contact, the scanning duration for each point is 8–13 ms but was strictly kept the same in control groups. None of the scanning images were filtered. For the control feedback parameters, the setpoint value, and proportional gain (PGain)/integrator gain (IGain) were mainly adjusted. The setpoint determines the amount of error signal changes in the approaching process. It defines a threshold of a state that could be regarded as in or out of contact. It is correlated to the force strength applied on the sample surface by the tip. The typical value of setpoint value for this device in scanning silicon wafer is 0.3, which should be decreased accordingly as the sample modulus decreases. The PGain is used to amplify the error signal, and IGain sets the integration time of the scanner. Higher PGain/IGain increases applied force strength, approaching speed and imaging resolution with significant increases in noise level and overshoots. In each comparison test, these parameters were set the same for applied tips. For silicon microgrid imaging, scanning parameters in the controlling software were set as follows: setpoint (0.3); PGain (0.1); IGain (1.8). For all the PDMS imagings, the gain parameters were fixed as PGain (0.5) and IGain (1.8), while the setpoint was tunable. The setpoint value was maintained at 0.35–0.4 for the spiral pattern scanning in Fig. 4b, while it was decreased to 0.1–0.13 to image the flower pattern in Fig. 4g. To compare the imaging difference using different CMA tips in Fig. 4i, the setpoint was controlled within a moderate range of 0.25–0.3. For biosample imaging, the PGain for both the CMA tip and the commercial tip was specified to 0.3 to enhance the mechanical interaction of tip–sample purposely, ten times the standard range of 0.01–0.03 for soft samples. The setpoint was kept at 0.3.

The commercial probe employed in the imaging of silicon calibrating grating has a vertically cantilevered tip (CAFP, Nanonics Imaging Ltd.) with an apex diameter of 20 nm and a nominal spring constant of 2600 N m$^{-1}$. The resonance frequency is 33.08 kHz, and the quality factor is 1795. The commercial probe employed in biosample imaging has a cantilevered NSOM optical fiber tip designed for cell imaging. The tip apex features a radius of 100 nm, a resonance frequency of 37.48 kHz and a quality factor of 920. The mechanical parameters of relevant CMA probes are listed in Supplementary Table 2.

The silicon calibrating grating (HS-100MG) is purchased from BudgetSensors company. The imaging of PDMS patterns was performed on the same sample and quartz slide substrate underneath. This PDMS (Sylgard 184 from Dow Corning Company) sample is a bulk polymer solid with a macroscopic size of $11.5 \times 10.5 \times 1.75$ mm$^3$, which was reversely molded from DLW 2D prints for the interfacial micropatterns (Supplementary Fig. 7a, b). To prepare the sample, the elastomer base and curing agent were mixed at a typical base/agent mass ratio of 10:1. The mixture was dipped on the 2D template covering the gratings and was then thoroughly degassed under vacuum to remove bubbles. The mold was carefully stripped off and sliced after 12 h solidification at 35 °C.

To prepare the fibroblasts, the normal human dermal fibroblasts (NHDF) from the Cell Bank of the Chinese Academy of Sciences (Shanghai, China) were maintained in CMAEM (high glucose, Gibco) medium and supplemented with 10% fetal calf serum (Sigma, USA), 100 U mL$^{-1}$ penicillin (Sigma, USA), and 100 mg mL$^{-1}$ streptomycin (Sigma, USA) at 37 °C with 5% CO$_2$ in a 95% humidified atmosphere. The cells were seeded on cover glasses and incubated at 37 °C for 12 h, which were then fixed with 4% paraformaldehyde solution for 10 min at room temperature. The sample was washed three times with phosphate-buffered saline (PBS) to remove cell debris caused by fixation.

To prepare the bacterial sample, *Shewanella oneidensis* MR-1 (ATCC 700550) were cultured in LB-medium (2.1 wt%) over night in a shaker flask (150 rpm, 37°C) and then centrifuged at 5000 rpm for 10 min, and subsequently fixed by 2.5 vol% glutaraldehyde solution at 4 °C for 2 h. To dehydrate the bacteria, the fixed cells were rinsed twice with deionized water and then resuspended in 25, 50, 75, 80, 95 vol% ethanol solution and absolute ethanol for 20 min subsequently. The MR-1 ethanol suspension was dipped onto a cleaned quartz substrate and dried at room temperature for imaging tests.

**Reactive ion etching process**. Tip etching was performed on Etchlab 200 from SENTECH Instruments GmbH. The experiment configurations were specified as follows: 35 sccm of O$_2$ flow rate, 10 sccm Ar flow rate, 8 Pa working pressure, and 50 w RF voltage. The etching duration was controlled for structural tuning.

**Measurement for boxplot statistics**. The statistics of step height in the boxplot of Fig. 3d are obtained by measuring and fitting the profiles of the same 30 step patterns in each tip-scanning image of Fig. 3a. The statistics of FWHM width and depth in the boxplot of Fig. 4f are obtained by measuring and Gaussian-fitting the profiles of the same 30 groove patterns in the height plots of Fig. 4b. Relevant data are provided as a Source Data file.

**Reporting summary**. Further information on research design is available in the Nature Research Reporting Summary linked to this article.

## Data availability

The experimental data supporting this study are available from the corresponding authors upon reasonable request. Source data are provided with this paper.

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

## Acknowledgements

This work was financially supported by National Key Research and Development Program of China (No. 2017YFA0700500), National Natural Science Foundation of China (No. 21902024 and 21327902), Natural Science Foundation of Jiangsu Province (No. BK20180408). We are grateful for in situ indentation tests performed by the Centre for Advanced Mechanics and Materials at Tsinghua University and technical support from Nanonics company.

## Author contributions

Z.Z.G., X.W.Z., Z.Y.L., and L.D.S. developed the idea. H.C.G. instructed manufacturing skills. L.D.S designed and fabricated the probes, performed numerical analysis, comparison experiments, characterizations, data interpretations and edited the layout of figures. Y.Z., Q.W.L., and N.C. prepared the biosamples for scanning. L.D.S. drafted the manuscript. Z.Z.G., X.W.Z., H.Y.C., H.C.G., D.Z., H.B.N., and X.J.L. revised the manuscript.

## Competing interests

The authors declare no competing interests.
