## [Peer Review File · Nature Communications]

Reviewer Comments

Reviewer #1 (Remarks to the Author):

The authors report on CMA probes demonstrating their use in AFM domain as new probes for imaging various samples (stiff and soft ones). Although, reasoning why they introduce new way of AFM probe production is relatively well described, it is difficult to judge whether CMA probes will be suitable. This should be better described. In my opinion, in the current form, the manuscript is not suitable for Nature Communications.

1. I have an impression that the authors somewhat misunderstand what the AFM probe. AFM probe (cantilever) contains two major part a tip (usually pyramid) and a level. Properties of the tip or more precisely the interaction between the atoms at the end of the tip and sample surface is the basis for a good quality image. Mechanical properties of the tip (their Young's modulus) rather weakly influence the image quality as compared to its surface properties. I did not get the clue why elastic modulus of the materials used to produce AFM probe will affect image quality.

2. It is true that the stiffness of the sample should be comparable to the stiffness of the tip. This refers to the lever spring constant (i.e. a cantilever spring constant). This is totally distinct properties than the material elastic modulus. When level deflect due to interacting force, cantilever spring constant enable to convert a deflection into force.

3. To demonstrate the quality of the CMA probes, I would compare their resonant frequencies, Q-factors that influence cantilever spring constants calibration, SEM images showing probe geometry before and after the imaging, for minimum 10 probes and compares this to commercial ones. Presented images show similar step height on silicon grating for all types of probes. PDMS seems to be better imaged by CMA probes but as compare to biological samples - it seems that this particular commercial probe was not of high quality.

4. As a commercial probes I would choose cantilevers with similar geometry i.e. radius of curvature of 2 nm not 20 nm. PDMS image can be better imaged with this probe. Biological samples - they are of typical images with any probe in dried conditions. Did you try to use your CMA probes in liquid conditions ? Maybe they will have better surface properties and improve quality of the images in this conditions.

5. Statistics: the number of CMA probes should be stated. Any number used in the text like 201 nm or 578.3 nm must be presented with statistically determined error or accuracy level.

6. Fig. 7 c & d - rainbow scale hides specific features present on a surface - these images should be presented either as white& black images or using a scale presented in fig.7 a (phase). Present visualization does not show an improved quality of new probes. Differences results from that fact that two different cells are presented. Does similar set point (load force) was applied to image these samples ? The fact that with commercial probes image change after 5 round may result from mismatching imaging conditions. The experimental part should be more precisely described.

Reviewer #2 (Remarks to the Author):

This paper reports a new idea of making AFM probe tips with controlled microstructural architecture. Applying the direct laser writing technology to program the structure of AFM probe tips, authors demonstrate such tunable CMA tips covering the range from 10 MPa to 2 GPa. The main purpose of the authors developing CMA probe is to solve the modulus mismatching problem between the sample and probe.

However, the paper suffers from a very significant flaw. Based on the current format, it cannot theoretically or experimentally prove that the modulus mismatching problem can be solved by

changing the tip stiffness of the probe. Thus, the paper cannot be published in its current format on NCOMMS.

In the measurement process, the core function of the modulus matching principle is to make the force between the tip and sample interaction force within the appropriate range. For example, in an AFM system that uses optical lever as force detection method, the mean value of the force during an oscillation is $F = kA_0/2Q[1 - (A_{sp}/A_0)^2]^{0.5}$ in tapping mode. (Lozano, J. R.; Garcia, R. Phys. Rev. B 2009, 79, 014110). Thus, the force can be effectively controlled by changing the stiffness of the probe.

Changing the stiffness of the tip (as the authors have done) reduces the whole stiffness of the probe, but the stiffness of the cantilever does not change. Thus, in the measurement system for detecting the force of the probe (depending on the deflection of the cantilever beam), since the size and stiffness of the cantilever beam are not changed, the probe force sensitivity (unit: nN/mV) remains the same. This means that as long as the set point of amplitude voltage is the same, the average force applied to the cantilever beam independent of the stiffness of the tip. Theoretically speaking, the stiffness matching problem cannot be solved simply by tuning the stiffness of the tip. In my opinion, the contribution of this paper is not to solve the stiffness matching problem, but to add a buffer layer between the one prong of tuning fork and the sample through a soft CMA tip, which does not change the tip-sample interaction force at the steady state of each measuring point. However, the transient force can be reduced when the probe change from one measuring point to another. If the authors could focus on this point, this paper would be a good paper for a technical journal.

There are some other comments on the experiments of this paper.

1. The direction of applied force used to test mechanical performances of the CMA probes (Fig. 3 and supplementary Fig. 2) is perpendicular to the bottom, while in the experiment the direction of the tip-sample interaction force (Supplementary Figure 3c) is parallel to the bottom. Is it right to use the mechanical property of CMA probes obtained by test and FEM analysis represent the stiffness in the experiment (shear-force feedback mode)?

2. Image quality depends on many aspects, such as tip-sample interaction forces, tip shape and radius. Therefore, the authors should give SEM images and Q factors of the CMA probe and solid probe for PDMS imaging. In this experiment, it is better to use two CMA probes with different tip stiffnesses directly to ensure that the shape and radius of the tip end are consistent.

3. Normally, cantilevered probes use normal-force feedback in tuning fork AFM, while the CMA probes operate in the shear-force feedback mode. Do commercial cantilevered probes mentioned in this manuscript also works in the shear-force feedback mode? If not, is it appropriate to use the results of different feedback modes for comparison?

4. What is the imaging performance of the CMA probe in the normal-force feedback mode? If possible, compare it with the shear-force feedback mode.

5. Can the author give a detail guidance on choosing the CAM structure based on the different Young's modulus of the sample?

Reviewer #3 (Remarks to the Author):

The manuscript by Sun et al describes a method for using direct laser writing (DLW) to fabricate atomic force microscope (AFM) probes. Unfortunately, I was not able to gleam a single piece of use or novelty from the manuscript. In short, they work very hard to design a mechanical structure, realize it using DLW, but fail to show that it provides any advantage over conventional approaches. Further, the manuscript is full of needless hyperbole and poor grammar. Finally, their representation of the prior literature is incorrect and ultimately ironic. Specifically, while they acknowledge that prior work has been done using DLW to realize AFM probes, they claim that these approaches have "essentially no competitive advantages over conventional ones." The references they cite provide counter examples of this claim including Ref 27 – probes with 10x higher bandwidth than conventional ones, Ref 29 – tailoring adhesion between the tip and substrate, and Ref 30 – tailored multi-frequency applications. The reason that this passage is ironic is that the present work provides no advantages over conventional approaches. Specifically,

they work hard building lattice structures that end up being extremely stiff compared with conventional probes (they demonstrate 13-386 N/m spring constants while conventional probes are 0.03 to 40 N/m). Thus, even though the structures they write appear complex, they provide nothing that conventional probes cannot provide. If the manuscript was substantially edited to remove the needlessly self-aggrandizing language, the figures were improved to actually make them legible, and the references to prior literature were improved, I can see this being considered in a much more specialized journal such as Ultramicroscopy or Nanotechnology. However, I cannot support publication in its present form as there are no technological innovations presented.

Author's Response to Reviewer #1:

Response to the questions one by one

The authors report on CMA probes demonstrating their use in AFM domain as new probes for imaging various samples (stiff and soft ones). Although, reasoning why they introduce new way of AFM probe production is relatively well described, it is difficult to judge whether CMA probes will be suitable. This should be better described. In my opinion, in the current form, the manuscript is not suitable for Nature Communications.

Response: We would like to thank the reviewer for taking his precious time to evaluate our work and helping us improve its quality. After receiving the comments, we carefully checked the whole manuscript and overwritten it. The new version of our manuscript includes updates in proofs, logics and languages. Specifically, in order to demonstrate the selection and contribution from the CMA probe, the comparison with a typical solid probe has been performed in dynamic impact simulations as well as practical imaging experiments on soft samples. Detailed discussions have been amended in relevant parts of the new manuscript, which we hope could answer the reviewer's questions.

1. I have an impression that the authors somewhat misunderstand what the AFM probe. AFM probe (cantilever) contains two major part a tip (usually pyramid) and a level. Properties of the tip or more precisely the interaction between the atoms at the end of the tip and sample surface is the basis for a good quality image. Mechanical properties of the tip (their Young's modulus) rather weakly influence the image quality as compared to its surface properties. I

did not get the clue why elastic modulus of the materials used to produce AFM probe will affect image quality.

Response:

1. We have to apologize for our careless use of the term 'AFM'. As the reviewer has commented, the common configuration for a typical AFM is based on the feedback of an optical lever. The exerted force could be evaluated measuring deflection when probe approaches the surface. However, there are various variants of AFM devices. In our configuration, the setup of an optical lever is canceled, which means the accurate control of impact force is disabled. Instead, the probe vibrates at a resonant frequency of tuning fork (manuscript Fig.1c), and a certain amount change (i.e., setpoint) of vibration amplitude or phase can be used as a criterion of contact. In this case, the impact force is not actually atomic force, and mechanical properties of the tip would induce a difference in impact stress level that affects the imaging quality. This could be experimentally demonstrated by the imaging difference of soft samples using the CMA probe and solid probe, as illustrated in manuscript Fig.4. Based on the height measurement, we could further estimate that the stress exerted by the CMA probe is about 13.8% compared to the solid probe. We could guarantee the authenticity of the scanning results. In fact, at first, we didn't mean to improve image quality at all when we first tried this cellular probe (i.e., CMA probe). This probe was intended to test the basic scan capability along with other designs but surprisingly exhibits a gift in imaging optimization. Other probes, including 3D-printed solid probes and commercial probes, were tested but failed to reach the same imaging quality no matter how we tuned the feedback parameters (setpoint, Proportional Gain, Integrator Gain). Judging from the evidence, we think the imaging optimization surely comes from the structure related mechanical properties.

2. It is obvious that the application of the term 'AFM' is inappropriate because our experiments are based on the shear force imaging mode of SPM. Therefore, the precedent version of the title has been replaced, becoming '3D-printed cellular probes for scanning probe microscopy (SPM)'. The relevant discussions, the device configurations and the parameters are supplemented in detail in the revised edition.

2. It is true that the stiffness of the sample should be comparable to the stiffness of the tip. This refers to the lever spring constant (i.e., a cantilever spring constant). This is totally distinct properties than the material elastic modulus. When level deflect due to interacting force, cantilever spring constant enable to convert a deflection into force.

Response: The reviewer is right about this problem. In the revised paper, relevant characterizations of stiffness are unified with spring constants instead of modulus.

3. To demonstrate the quality of the CMA probes, I would compare their resonant frequencies, Q-factors that influence cantilever spring constants calibration, SEM images showing probe geometry before and after the imaging, for minimum 10 probes and compares this to commercial ones. Presented images show similar step height on silicon grating for all types of probes. PDMS seems to be better imaged by CMA probes but as compare to biological samples - it seems that this particular commercial probe was not of high quality.

Response:

1. The reviewer requests specific measurements on the mechanical parameters as well as wear tests. However, the amount of probes for testing requires laborious operations, which we don't think is very necessary. First, as we described in the manuscript, the assembly process of CMA probes for SPM includes 3D printing cellular body on the fiber end face,

cutting the fiber into 3~5 mm pieces with structure locating at one end and attaching it to one arm of the tuning fork using UV-adhesive. If we have to choose SEM to characterize the probes before and after scanning, we have to detach the fiber after each scan and place it in the SEM chamber for observation. However, it's very difficult to detach it without breaking the structure. It's true we could place the whole tip-mount into the SEM chamber, but considering we only have 4 available tuning forks for tests, it means at least 6 times of repeated SEM experiments are needed. Instead, for comparison, we measured the mechanical parameters, including resonant frequencies and Q-factors of the available four tuning forks before and after the assembly process. For the wear tests, we use an inverted microscope with a 60x objective to extract the structural shapes after repetitive scans (40 times for each probe). Relevant data and discussions are appended in Supplementary Table 1, Supplementary Fig.10 and Supplementary Discussion.

2. The reviewer thought the scans images of silicon microgrids by CMA probe and commercial probes are similar. It is true. By scanning the hard silicon sample, we were intended to explore two things. The one is to test the essential scanning capability of the 3D-printed CMA probe, and this was demonstrated by the consistent measurement of step height as the reviewer commented. The other goal is to explore the specific influence on imaging by mismatched stiffnesses of probe-sample. In manuscript Fig.3, the instability of the softer CMA probe during scanning degrades final image quality, which demonstrates the necessity of stiffness-matching guideline in structural design. The relatively low quality of biosample images by commercial probe does not come from the low quality of the probe itself (resonance frequency of 37.48 kHz with a quality factor of 920). For biosample imaging, the value of PGain/IGain setting of both the CMA probe and the commercial probe was ten times increased from the recommended 0.01-0.03 for soft samples to 0.3 to enhance the mechanical interaction of sample-probe purposely.

Therefore, the relatively good image quality by the CMA probe exhibited a competitive performance compared with the commercial products.

4. As a commercial probes I would choose cantilevers with similar geometry i.e., radius of curvature of 2 nm not 20 nm. PDMS image can be better imaged with this probe. Biological samples - they are of typical images with any probe in dried conditions. Did you try to use your CMA probes in liquid conditions ? Maybe they will have better surface properties and improve quality of the images in this conditions.

Response:

1. In terms of image quality, the tip size of the probe is surely an important factor but not the only one. The presented image difference of PDMS doesn't come from the tip size. In fact, the probes we used in this comparison experiment were both fabricated by 3D printing with the apex composed of one single voxel. The tips were controlled in the same size as given by Supplementary Discussion and Supplementary Fig. 6. Therefore, the image difference comes from the mechanical properties of the structures in this condition. By simply selecting a probe with a smaller tip, sample deformation induced by impact stress could not be reduced directly. Thus, in our opinion, the image will not be significantly optimized with this probe.
2. We have tried to perform the scan in a liquid environment but failed. The feedback is unstable when exposed to evaporated water vapor from aqueous solution, and it is impossible for us even to get a correct and stable contact signal (Attached video file of 'Cover letter movie 1' below shows the unstable error value when tips are moving towards liquid interface). Suggested by the engineers, we are now trying to use glycerol to replace the water as liquid media for scanning. However, in this research, it is unnecessary to test the scans in liquid environments. First, the liquid media will damp the impact

energy, which makes the structure property not the only variable in imaging optimization. Second, with liquid filling in the pores of cellular structure, the compressive response will change. The simulation will lose the reference value. Therefore, in order to keep overall logical consistency in the context. We would prefer performing the experiment in a dry environment.

C o ver letter
m o vic 1-unstable

5. Statistics: the number of CMA probes should be stated. Any number used in the text like 201 nm or 578.3 nm must be presented with statistically determined error or accuracy level.

Response: The statistics have been revised in accordance with the reviewer concerns

6. Fig. 7 c & d - rainbow scale hides specific features present on a surface - these images should be presented either as white& black images or using a scale presented in fig.7 a (phase). Present visualization does not show an improved quality of new probes. Differences results from that fact that two different cells are presented. Does similar set point (load force) was applied to image these samples ? The fact that with commercial probes image change after 5 round may result from mismatching imaging conditions. The experimental part should be more precisely described.

Response:

1. In accordance with suggestions by the reviewer, we have updated the height images of cells as well as a new false-color scheme in Fig.5. Corresponding phase details are also supplemented. The repetitive scan tests were performed on the same cell species (Shewanella MR-1 bacteria cells). Since the cell surface deformed by the probes during

repetitive scans is hard to recover, it is inappropriate to perform the comparison experiments on the same cell.

3. Similar setpoint was applied to image these samples, and experimental details are recorded in the Method part in the revised manuscript.

Author's Response to Reviewer #2:

This paper reports a new idea of making AFM probe tips with controlled microstructural architecture. Applying the direct laser writing technology to program the structure of AFM probe tips, authors demonstrate such tunable CMA tips covering the range from 10 MPa to 2 GPa. The main purpose of the authors developing CMA probe is to solve the modulus mismatching problem between the sample and probe.

However, the paper suffers from a very significant flaw. Based on the current format, it cannot theoretically or experimentally prove that the modulus mismatching problem can be solved by changing the tip stiffness of the probe. Thus, the paper cannot be published in its current format on NCOMMS.

In the measurement process, the core function of the modulus matching principle is to make the force between the tip and sample interaction force within the appropriate range. For example, in an AFM system that uses optical lever as force detection method, the mean value of the force during an oscillation is $F = kA_0/2Q[1 - (A_{sp}/A_0)^2]^{0.5}$ in tapping mode. (Lozano, J. R.; Garcia, R. Phys. Rev. B 2009, 79, 014110). Thus, the force can be effectively controlled by changing the stiffness of the probe.

Changing the stiffness of the tip (as the authors have done) reduces the whole stiffness of the probe, but the stiffness of the cantilever does not change. Thus, in the measurement system for detecting the force of the probe (depending on the deflection of the cantilever beam), since the size and stiffness of the cantilever beam are not changed, the probe force sensitivity (unit: nN /mV) remains the same. This means that as long as the set point of amplitude voltage is the same, the average force applied to the cantilever beam independent of the stiffness of the tip.

Theoretically speaking, the stiffness matching problem cannot be solved simply by tuning the stiffness of the tip. In my opinion, the contribution of this paper is not to solve the stiffness matching problem, but to add a buffer layer between the one prong of tuning fork and the sample through a soft CMA tip, which does not change the tip-sample interaction force at the steady state of each measuring point. However, the transient force can be reduced when the probe change from one measuring point to another. If the authors could focus on this point, this paper would be a good paper for a technical journal.

Response: We would like to thank the reviewer particularly. His advice significantly promotes the further improvement of our works and enlightens us with a new perspective on this research. Here are our responses.

1. First, the scanning device we use is not based on typical optical lever feedback, with which the force applied to the surface should indeed have been controlled effectively. Thus, the model discussed and provided by the reviewer in the reference is not very appropriate for the determination of our case. Moreover, the feedback of our devices is based on the measurement of the motion changes of the tuning fork (e.g., the phase or amplitude). The setpoint value accordingly represents the maximum threshold of that change, which determines the current state as in contact if the value is exceeded. Therefore, the setpoint represents motion sensitivity instead of direct force sensitivity, and

a similar setpoint value doesn't mean a similar average force functioning in the whole impact process.

2. Analysis of the impact process is supplemented in support information, which is directly duplicated here to clarify the mechanism of stress mitigation based on CMA material during a specific impact process.

Supplementary Figure 7 | Schematic of the probe impact. The process is composed of four steps (i.e., approaching, approached, responded and stopped). The red indicates influence factors in the corresponding steps, and the blue shows the control methods.

'To better clarify what role CMA material may have in the scanning, the impact process is divided into four steps, as illustrated in Supplementary Fig. 7. After the approaching step is initiated, in the second step, the probe will subsequently contact the surface determined by preset setpoint value. However, the scanner would not stop immediately by the time it receives the in-contact signal. Specifically, the integrator gain parameter sets the time for the scanner to respond for any change of the error signal of setpoint. During this short period

between step 2 and step 3, the scanner would keep moving downward until the feedback is responded, and the probe is then decelerated until it finally stopped in step 4. Here, in addition to the necessary response time required by the hardware settings, feedback errors, including signal hysteresis and dysfunction, would significantly extend this duration. The final deceleration is mainly controlled by the scanner parameter settings, the loading speed of probe and its mechanical property. Accordingly, the control feedback parameters (i.e., setpoint, PGain and IGain) and the application of energy-absorbing CMA material are two main methods to control the impact process. The setpoint determines the amount of error signal changes of a tuning fork (phase or amplitude) in the approaching process, which returns a specific contact state, directly correlating to the force strength applied on the surface and indirectly affecting approaching speed combined with the tuning of gain parameters. Increased setpoint and gain parameters would increase the approaching speed of probe and thus enhance the contact interaction. Hence, parameter control is functioning in all four steps, which are strictly kept the same in the comparison experiments. Even though the contact force in step 1-2 is independent of probe selections by designating the equal value of setpoint, the positive buffer effect provided by the CMA probe still works effectively in the rest steps. The cellular lattices serving as sacrificial layers to sustain part of the displacement through the structural buckling during response and deceleration period rapidly decelerate the probe by absorbing its kinetic energy, which finally contributes to the mitigation of impact as well as surface deformation.'

Therefore, in the response and deceleration process, the average forces are no longer independent of probe mechanical properties since probe bulking will determine impulsive energy-absorbing performance and therefore reducing the sample deformation. Finite element analysis simulation on the dynamic impact process is supplemented in updated manuscript

Fig.2 and clearly clarified the change of sample surface and corresponding stress. Besides, deducing from the practical scanning results on PDMS patterns presented in Fig.4, the image difference is surely related to the structural properties of the cellular construction. If the average force is maintaining the same in the whole impact process for both probes, the optimization would not exist.

3. The assumption of the buffer layer effect is accepted with our sincere gratitude after thoroughly reviewing related references and serious discussions with experts in this area. In the revised version, the energy-absorbing performance is characterized, and the corresponding theoretical system is perfected.

There are some other comments on the experiments of this paper.

1. The direction of applied force used to test mechanical performances of the CMA probes (Fig. 3 and supplementary Fig. 2) is perpendicular to the bottom, while in the experiment the direction of the tip-sample interaction force (Supplementary Figure 3c) is parallel to the bottom. Is it right to use the mechanical property of CMA probes obtained by test and FEM analysis represent the stiffness in the experiment (shear-force feedback mode)?

Response:

Our scanning is based on the shear force imaging mode (Cover letter Fig 1a, b). However, in this case, the tip-sample interaction force, including both shear force and normal force. Even though the feedback comes from the vibration in shear direction, the amplitude is in nanoscale, and the mechanical interference with the sample surface by shear force could be neglected. When probe approaches the sample surface, the impact in the normal direction (Cover letter Fig. 1b) is the key problem that we are trying to optimize in this work. Here, the conventional manufacturing method of tuning cantilever could only control the shear force effectively but fails to buffer the normal impact. Therefore, the advantageous cellular construction fabricated

by 3D printing can solve this problem. Thus, in our opinion, it is still appropriate to use the mechanical property of CMA probes obtained by tests and FEM analysis to represent the stiffness in the experiment

2. Image quality depends on many aspects, such as tip-sample interaction forces, tip shape and radius. Therefore, the authors should give SEM images and Q factors of the CMA probe and solid probe for PDMS imaging. In this experiment, it is better to use two CMA probes with different tip stiffnesses directly to ensure that the shape and radius of the tip end are consistent.

Response: In accordance with the reviewer's request, the SEM pictures of both probes and details about their mechanical parameters are supplemented in support information (Supplementary Figure 6 and Table 2). The apex size of both probes is similar because both tips are only composed of one single voxel. The reason why we chose a 3D-printed solid probe as a control instead of using a commercial probe is to get rid of the influence of tip size. Since the presence of hierarchical cellular construction is the only difference, the imaging optimization has to be rooted therein. But we also supplement the image comparison between two CMA probes with different tip stiffnesses in Fig.4.

3. Normally, cantilevered probes use normal-force feedback in tuning fork AFM, while the CMA probes operate in the shear-force feedback mode. Do commercial cantilevered probes mentioned in this manuscript also works in the shear-force feedback mode? If not, is it appropriate to use the results of different feedback modes for comparison?

Cover letter Figure 1 | Tip-mount configurations for shear force and normal force imaging mode. (a, b) Tuning fork photograph and operation diagram based on shear force imaging mode. (c, d) Tuning fork photograph and operation diagram based on normal force imaging mode.

Response: All the probes we used in the experiments operate in the shear-force feedback mode due to its high degree of customization. As shown in manuscript Fig.1c, the fiber with the 3D-printed structure on the facet was reduced to 3~5 mm in length, which was subsequently bonded to one side of tuning fork. If normal-force feedback is applied, the attachment of fiber piece to the tuning fork would be very difficult, especially a perfect normal direction is needed (Cover letter Fig.1c). Besides, the location of tips under the microscope will be hard to confirm, which poses a challenge in practical scanning. Directly printing nanostructures to the tuning fork without using an optical fiber as media is not currently applicable because the chemicals used for resist developing would corrupt the

components of the tip-mount. Since the normal impact (Support Fig.1b, d) we were trying to optimize is independent of specific feedback modes, therefore we don't think it is necessary to perform the whole experiments again with normal force feedback.

4. What is the imaging performance of the CMA probe in the normal-force feedback mode?
If possible, compare it with the shear-force feedback mode.

Response: We have explained the question in the response of the 3rd comment.

5. Can the author give a detail guidance on choosing the CAM structure based on the different Young's modulus of the sample?

Response: In the selection of CMA probes for better buffer effect, we follow the stiffness-matching rules of probe-sample in specific cellular design. That is, with a given stiffness value of the sample, the lattice size of the CMA structure would be determined by FEM simulations to yield a similar stiffness. The stiffness-matching rules are deduced theoretically and experimentally in the revised manuscript.

Author's Response to Reviewer #3:

The manuscript by Sun et al describes a method for using direct laser writing (DLW) to fabricate atomic force microscope (AFM) probes. Unfortunately, I was not able to gleam a single piece of use or novelty from the manuscript. In short, they work very hard to design a mechanical structure, realize it using DLW, but fail to show that it provides any advantage over conventional approaches. Further, the manuscript is full of needless hyperbole and poor grammar. Finally, their representation of the prior literature is incorrect and ultimately ironic. Specifically, while they acknowledge that prior work has been done using DLW to realize AFM probes, they claim that these approaches have "no competitive advantages over

conventional ones.” The references they cite provide counter examples of this claim including Ref 27 – probes with 10x higher bandwidth than conventional ones, Ref 29 – tailoring adhesion between the tip and substrate, and Ref 30 – tailored multi-frequency applications. The reason that this passage is ironic is that the present work provides no advantages over conventional approaches. Specifically, they work hard building lattice structures that end up being extremely stiff compared with conventional probes (they demonstrate 13-386 N/m spring constants while conventional probes are 0.03 to 40 N/m). Thus, even though the structures they write appear complex, they provide nothing that conventional probes cannot provide. If the manuscript was substantially edited to remove the needlessly self-aggrandizing language, the figures were improved to actually make them legible, and the references to prior literature were improved, I can see this being considered in a much more specialized journal such as Ultramicroscopy or Nanotechnology. However, I cannot support publication in its present form as there are no technological innovations presented.

Response:

We thank the reviewer for the questions on the novelty and language. We apologize for our poor English skills, which poses a challenge to the comprehensive reading of the manuscript. We have seriously overwritten the manuscript with updated experimental data, figures, and languages. We removed the redundant and inappropriate remarks and corrected the errors of grammars and vocabulary. The manuscript has been edited by a native speaker before submission. We sincerely hope this version could meet the reviewer’s request as well as the publication standard. In addition, we would like to address the reviewer’s questions as follows. By writing ‘no competitive advantages over conventional ones’ in the original version, we didn’t mean to deny the novelty contributed by those articles. In fact, we are very grateful to have them as references, which instruct us in experiments design and implementation. We

intended to express that most SPM studies based on 3D-printed probes didn't fully take advantages of the 3D structural programmability, without which, as far as we're concerned, the structure related properties are disabled and the presented novelty by the article could not sufficiently make them as competitive as the commercial alternatives:

- In Ref 27 (Ref 8 in the current manuscript version, *Small* 2018, 14, 1800162), the novelty is stated by 'Specifically, the low Q of polymer probes allows them to scan with an order of magnitude higher bandwidth than conventional probes'. In fact, the decrease of the Q factor is a by-product of the manufacturing process. Accurate control of Q by 3D printing is not the topic of this article. Alternatively, in commercial applications, the Q can be decreased by attaching additional mass to the cantilever without the application of 3D-printing. Besides, the probe employed a typical cone shape, and the only benefit from 3D printing is to control the size of the fabricated cantilever in order to tune the vibrational resonance frequency and stiffness, which is also the typical design and scheme applied in commercial design. Therefore, we think the significance provided by the manufacturing process, the design and the novelty are not so remarkable to distinguish itself from conventional methods.
- In Ref 29 (deleted in the current manuscript version, *Nanotechnology*, 2016, 27(15): 155702), the tailoring adhesion between the tip and substrate is actually the result of hydrophobic PFPE-based polyfunctional photoresist, and in fact, the hydrophobicity could be realized by long term silanization in room temperature. Even though the author claimed 'Typically, tips are manufactured in hard inorganic silicon-based materials by standard microfabrication technologies that are limited in the customization of the tip design.', the article doesn't appear to demonstrate the necessity of customized design by 3D printing. Therefore, we think the advantage of 3D printing is underused.

- In Ref 30 (Ref 35 in the current manuscript version, Applied Physics Letters, 2016, 109(6): 063101), the article tries to explore the feasibility of a 3D-printed probe for AFM. We have to admit the mechanical tuning of the cantilever is really a brilliant idea by 3D printing rebar structures to cantilevers for multifrequency AFM. This can hardly be done by conventional manufacturing methods. We are very sorry for our careless and inappropriate comments in the original manuscript version.

In our research, 3D printing was employed to fabricate the cellular structures. First, the tunable stiffness properties could be afforded by programming lattice size, which contributes to the cushion performance in each impact during scanning. Therefore, the manufacturing process of the cellular solids could not be replaced by conventional methods, which confirms the necessity of 3D printing. Second, as for the necessity of cellular construction, the bulking of the compressible structure mitigates the impulsive energy that should have exerted on the sample surface causing severe deformation. It could not be approached by typical solid probe design. Last but not least, the commercial probe could indeed achieve a lower stiffness by modifying the geometrical size of a cantilever. However, considering that the shear-force imaging mode was employed in our SPM tests, in which case the probe vibrates in a parallel direction to the sample surface, geometrical tuning of the cantilever is not an effective way to optimize the collision in the normal direction. It is necessary to build a buffer layer between the sample and the probe. As far as we know, this is beyond the reach of commercial schemes. Thus, we think our work could provide some advantage over conventional methods.

As a short conclusion of the novelty presented by our work, it updates the current SPM studies based on 3D-printed probes by perfectly taking advantage of its structural programming ability and offers a novel probe design with tunable mechanical property. Second, this research provides an advantageous energy-absorbing application of cellular mechanical metamaterial in impact mitigation.

We hope our explanation could answer the reviewer's questions. We thank the reviewer for spending his valuable time evaluating our work and giving precious suggestions. It really helps us further improve the quality of the manuscript.

Reviewers' comments:

Reviewer #1 (Remarks to the Author):

The authors showed a substantial improvements of the manuscript in comparison with previous version. All issues raised by me were addressed in a satisfactory way. Therefore, my recommendation is to accept.

Reviewer #2 (Remarks to the Author):

I am pleased to see that the revised version has been changed to focus on the energy-absorbing behavior of CMA materials used as the tuning fork AFM tip during scanning. The manuscript details why the CMA tip has energy buffering and how it relates to imaging quality. Although the tuning fork AFM makes the role of the CAM material energy absorption prominent due to its large stiffness, the CAM tip also provides a new perspective for cantilever AFM imaging. Based on these grounds, I recommend this manuscript for publication in Nature Communications after the following comments have been addressed.

1. Due to the probe composed of a tuning fork and a tip, it is not rigorous to mention that the probe is made of CMA material in many places in the manuscript. In fact, authors just use CMA material to construct the tip.
2. Scanning probe microscope (SPM) is a generic term covering a group of techniques, such as scanning tunneling microscope (STM), atomic force microscope (AFM), and scanning ion conduction microscope (SICM). Obviously, CMA materials are not suitable for tips of STM or SICM. Therefore, it is more appropriate to limit its application area to AFM.
3. Which compression stage is the CMA tip in during scanning? If it is in the first elastic regime, how to interpret the defined absorbed energy (W_{absorbed}) includes the plastic stage (Fig. 2a). If it is in the plastic stage, the deformation of the CMA tip cannot be completely recovered at this stage, will it affect the imaging quality? And at this stage, the changes in strain and stress are unstable (Figure 2b). Will this affect the imaging quality?
4. The authors attribute the different image qualities of Fig. 4b and c to the material inhomogeneity at the nanoscale. Is it possible that the morphology of the substrate may cause non-uniformity in the stiffness of the material? From Fig. 4f and Eq. 1-8 in supplementary information, it can be estimated that L2 is not much larger than (L1-L2), otherwise the same stress will lead to the same indentation of the same material. What is the thickness of the PDMS sample (L2 in Supplementary Fig. 8)?
5. Since the inhomogeneity of PDMS at nanoscale is not particularly obvious, it is not very powerful to prove that the solid tip highlights the stiffness difference of the sample rather than reflecting the real topography. Can the authors use block copolymer, whose different blocks have different stiffnesses at the nanoscale, as a test sample to further prove this point?

Reviewer #3 (Remarks to the Author):

While the manuscript is improved in terms of clarity, the problem solved by the technological innovation is very minor and better suited for a more specialized journal such as Ultramicroscopy. I appreciate the shift of focus from tuning elastic parameters to energy absorption, but it is not clear from the data how important or useful of an advance this is. Further, even if this solves a problem for this mode of scanning probe microscopy, this is itself a highly specialized and uncommon variant of atomic force microscopy. While I do not recommend publication in Nature Communications, the authors should consider the following changes before moving forward elsewhere.

- 1) While I understand the title was changed to attempt to narrow the scope, it does the opposite. Scanning probe microscopy is a more general term than atomic force microscopy. I recommend making the title reflect the actual content of the paper, such as "3D printed cellular probes for gentle contact in shear-mode atomic force microscopy."

2) Shear force feedback is presented as a feature that is unique to tuning forks, which seems to ignore the great body of work studying lateral force microscopy using optical levers.

3) At a high level, I understand the main virtue of the lattice being to buckle into a normal-force-limited state when the probe is put into contact for shear mode imaging. This leads to two major points (A) an image should be provided in the manuscript (Fig 1) that shows the geometry during imaging and the expected configuration of the lattice during imaging and (B) this concept should be explored by imaging the same sample using multiple set points to evaluate how this buckling affects image quality.

4) The buckling of the lattice during contact likely leads to a decrease in the lateral resolution, which seems evident in Figures 3 and 4. It is not clear this tradeoff will be worth it for samples that are not damaged during imaging.

5) Given that the polymer tip itself is not very hard (speaking here of the material property hardness as opposed to using the term hard to mean stiff as is done in the manuscript), it is unlikely it will damage any samples besides polymers and biological materials. Given the low lateral resolution (in part due to the large tip diameter), it is not clear this imaging approach actually brings anything to the table that is not already possible using other modalities of scanning probe such as tapping mode imaging using standard optical lever based approaches.

6) It is worth emphasizing for readers that this probe would not work in normal mode as the non-linear nature of the force-displacement curve would make it impossible to identify the sample height. This is a major limitation of this manuscript.

Author's Response to Reviewer #1:

The authors showed a substantial improvements of the manuscript in comparison with previous version. All issues raised by me were addressed in a satisfactory way. Therefore, my recommendation is to accept.

Response: We thank the reviewer for the encouraging comments.

Author's Response to Reviewer #2:

I am pleased to see that the revised version has been changed to focus on the energy-absorbing behavior of CMA materials used as the tuning fork AFM tip during scanning. The manuscript details why the CMA tip has energy buffering and how it relates to imaging quality. Although the tuning fork AFM makes the role of the CAM material energy absorption prominent due to its large stiffness, the CAM tip also provides a new perspective for cantilever AFM imaging. Based on these grounds, I recommend this manuscript for publication in Nature Communications after the following comments have been addressed.

1. Due to the probe composed of a tuning fork and a tip, it is not rigorous to mention that the probe is made of CMA material in many places in the manuscript. In fact, authors just use CMA material to construct the tip.

Response: We thank the reviewer for this precious advice and apologize for our negligence. To be more precise and rigorous as the reviewer suggested, we now replace the word 'probe' with 'tip' to describe the role of CMA structure (i.e., we now use 'CMA tip' instead of 'CMA probe'). The word 'probe' is defined as the integration of all micro-components of the scanning media, including a typical tip mount, a cantilevered tuning fork, an optical fiber and a tip constructed

by CMA material. Therefore, we would like to replace the precedent title (i.e., '3D-printed cellular probes for scanning probe microscopy (SPM)') accordingly, becoming '3D-printed cellular tips for scanning probe microscopy'. All relevant parts have been correspondingly revised and marked.

2. Scanning probe microscope (SPM) is a generic term covering a group of techniques, such as scanning tunneling microscope (STM), atomic force microscope (AFM), and scanning ion conduction microscope (SICM). Obviously, CMA materials are not suitable for tips of STM or SICM. Therefore, it is more appropriate to limit its application area to AFM.

Response:

We fully understand the reviewer's concerns about this issue. We have four reasons for the use of the term 'SPM' instead of 'AFM'. First, the applied scanning method is indeed based on a scanning probe system, which neither shares the same mechanism as typical AFM setup where the optical reflection is generally employed for scanning feedback and calibration nor consistently maintains the mechanical interactions to a molecular lever between few atoms within the van der Waals' contact. Therefore, it is rigorous not to use the term 'AFM' in this study. Second, however, the scanning tests imply application potentials in versatile scanning probe techniques including AFM, TERS or NSOM. Therefore, the selection of the generic term 'SPM' would be better, indicating a universal research benefit. Third, in our opinion, if the term 'SPM' is not suitably used just because the application field of CMA material could not cover all the techniques of SPM, then the use of 'AFM' is also inappropriate since AFM can be categorized into many branches with surely lots of prefixes according to different functionalities, contact modes and feedback modes. Conversely, SPM is a general concept of scanning microscopy, which can be and in fact has been widely used in both industry and academia. For example, the scanning device we employed in the study is named SPM

(Multiview 4000 from Nanonics Ltd.) which is used to commercially underline the great application potentials and functionalities even though not all SPM techniques are integrated into the device such as STM. Academically, a recently published paper entitled ‘3D-Printed Scanning-Probe Microscopes with Integrated Optical Actuation and Read-Out’ (Small, 2019, 1904695) mainly focused on updating near-field optical microscopy using 3D printed tips, which, however, does not impede the selection of SPM as the keywords even though not all SPM technical branches were realizable using this method. Last but not least, the CMA material is unrestricted and extendable in the application areas of SPM technologies where mechanical interaction exists. Instead of constructing the whole tip as was proposed in our work, it can be flexibly designed serving as tiny mechanical components at the connective joints of the tips according to the specific energy-absorbing need of versatile SPM technologies. Therefore, the statement of SPM in our manuscript is logically suitable in both narrow and broad senses, and we do not think it is crucial to change the term of SPM to AFM.

3. Which compression stage is the CMA tip in during scanning? If it is in the first elastic regime, how to interpret the defined absorbed energy (W_{absorbed}) includes the plastic stage (Fig. 2a). If it is in the plastic stage, the deformation of the CMA tip cannot be completely recovered at this stage, will it affect the imaging quality? And at this stage, the changes in strain and stress are unstable (Figure 2b). Will this affect the imaging quality?

Response:

1. The compression stage of the CMA tip is mostly distributed in the elastic regime during the scanning. This can be demonstrated by the wear tests presented in Supplementary Fig 13 (shown below), in which the structures almost completely recovered to its original shapes even after tens of scanning tests. The second evidence is from the section ‘Estimation of

stress level by CMA and solid tips' appended in the supplementary discussion, where the stress level can be determined from the measured height using this formula (equation 7 in supplementary information):

$$\sigma = M * \left(1 - \frac{\Delta h'}{\Delta h}\right)$$

Substituting for the parameters with measured data of 83.5 ± 6.0 nm, 85.9 ± 3.1 nm and 2.6 MPa respectively corresponding to $\Delta h'$, Δh and M, the average stress level of CMA tip ($a = 5 \mu\text{m}$) is about 0.0726 MPa. Given that the plateau stress value is above 0.1 MPa, the current compression stage should be within the elastic regime.

Supplementary Fig.13 Durability test of CMA tips. Four CMA tips (Case 1-4) were employed for the wear tests by repeated scans (40 times) on PDMS mold.

Even though the elastic regime is an ideal working condition since the tips could recover entirely from the deformation and thus extending its lifetime, as we have stated in the main manuscript, occasional overstressed situations were found especially when scanning steep-slope patterns or working in a super long-term scanning process, some of which would

exceed the peak stress leading to inevitable plastic deformation and make the tip gradually worn out in weeks. For one thing, this is the very reason that 3D printing, as well as an optical fiber, is strongly proposed for a customizable and deposable purpose, for another, in terms of energy-absorbing characterization, the worst compression condition, i.e., plastic deformation, should be taken into account by scaling the strain range from elastic regime up to plastic deformation stage.

2. The reviewer wonders why defined absorbed energy includes the plastic stage when the tip is elastically compressed. In the manuscript, the absorbed energy (equation 2 in manuscript) can be denoted by

$$W_{absorbed} = \int_0^{\varepsilon_{tr}} \sigma(\varepsilon) d\varepsilon$$

We have to apologize that there is a mistake in this formula and ε_{tr} should be replaced with a variable ε , which distributed in the range from 0 to ε_{tr} :

$$W_{absorbed} = \int_0^{\varepsilon} \sigma(\varepsilon) d\varepsilon \quad \varepsilon \in [0, \varepsilon_{tr}]$$

The formula for absorbed energy calculation should be an indefinite integral as a function of variable strain. Therefore, the mechanical behavior in the plastic regime is not taken into account in the energy calculation of an elastically compressed tip by correctly substituting the strain variable with value in the corresponding stage. Notably, the energy absorption diagram was built in Fig. 2d relating the absorbed energy to the stress level at the corresponding compression state, which underlines the quantitative changes of energy as a function of stress instead of separate discussions on specific compression stages.

3. The reviewer wondered about the possible problems caused by tip working on a plastic stage. Honestly, it is very difficult to identify the plastic deformation stage solely from the scanned images. Therefore, image degradation by CMA tip, if there is any, can not be directly attributed to the compression condition, in which way we can not affirm that plastic compression will certainly influence image quality. Instead, based on our experimental and

experiential knowledge, most tips are working on an elastic stage, and no significant evidence has been found that possible plastic compression would pose an instability in practical scan and affect the image quality. Interestingly, even after weeks' usage, the tips remained robust surpassing commercial alternatives.

4. The authors attribute the different image qualities of Fig. 4b and c to the material inhomogeneity at the nanoscale. Is it possible that the morphology of the substrate may cause non-uniformity in the stiffness of the material? From Fig. 4f and Eq. 1-8 in supplementary information, it can be estimated that L_2 is not much larger than (L_1-L_2) , otherwise the same stress will lead to the same indentation of the same material. What is the thickness of the PDMS sample (L_2 in Supplementary Fig. 8)?

Response:

1. The substrate morphology will cause non-uniformity in material stiffness, which is precisely what we have expressed in the manuscript. In fact, not only the sample nanolandscape but also the PDMS polymerization in nanoscale will randomly lead to minor differences in material stiffness distribution. In a word, the mentioned 'material inhomogeneity' can be understood as an interfacial difference in stiffness caused by material composition and morphology.
2. Here, we would like to explain further about the relations between imaging difference and the interfacial stiffness inhomogeneity. In the manuscript, the quantitative measurements on the sample surfaces and the groove heights were proposed as a part of the characterizations of imaging quality improved by CMA tips. As was discussed in the supplementary discussion titled 'Estimation of stress level by CMA and solid tips', the difference in groove height measurements could be attributed to the different stress level,

which proved that higher stress level exerted by solid tips corresponds to smaller height value in the measurement since the micropatterns were tightly pressed against the ground. However, apparently, it could not account for the different roughness measurements of the surface since solid tips seem to have a higher roughness value. The only reason has to be rooted in the stiffness difference of the sample surface. Specifically, in the height plot of Fig. 4b, c, the imaging deviation of the area indicated by the white arrow can be readily discerned. The light-colored contours given by the height plot of Fig. 4c, i correspond to a stiffer region distinct from the surrounding neighborhood, which rises as a highland after tip interaction. Briefly, interfacial stiffness inhomogeneity and morphological features of nanopatterns are key factors that affect final imaging results. Higher stress level exerted by tips will aggravate this influence by increasing morphological fluctuation of a flat surface but conversely decreasing the height contrast of stage pattern, which degrades the imaging accuracy and calls for the impact mitigation from the CMA tips. We particularly thank the reviewer for this comment since it enlightens us that interfacial stiffness difference should be taken into account when calculating the stress level from the measured data of groove height. The relevant approximation of the stress level in supplementary information has been updated.

3. We do not quite understand the reviewer's estimation of the quantitative relations between L_2 and (L_1-L_2) . In our opinion, it is not convincing to infer that L_2 is not much larger than (L_1-L_2) from the stress-strain relations. Assuming that we are adding infinite flat blocks under the model given in Supplementary Figure 8 to support the stage pattern (Cover letter Fig.1), the value of (L_1-L_2) will remain constant while L_2 can be increased to an infinite value. The algorithm presented in Eq. 1-8 of supplementary information remains unchanged after stacking the additional layers underneath since the infinite underlying layers of the higher and lower stage will contribute to the same indentation displacement (i.e., Δl_3 in

Cover letter Fig.1), which will be subsequently canceled in the calculation of the measured height difference using Eq. (5)

$$\Delta h' = (L_1 - \Delta l_1) - (L_2 - \Delta l_2)$$

In brief, the stage height (i.e., $L_1 - L_2$, Δh or $\Delta h'$) is a relative physical quantity that is independent of sample thickness (i.e., L_1 or L_2). It is meaningless to discuss the specific thickness of PDMS samples since it does not affect the tip-sample interaction, interfacial deformation, or scanning imaging.

Cover letter Fig.1 Updated compression model by adding additional layers underneath.

5. Since the inhomogeneity of PDMS at nanoscale is not particularly obvious, it is not very powerful to prove that the solid tip highlights the stiffness difference of the sample rather than reflecting the real topography. Can the authors use block copolymer, whose different blocks have different stiffnesses at the nanoscale, as a test sample to further prove this point?

Response:

1. We have explained the reason that we attributed the difference of measured surface roughness to the divergence of sample stiffness distribution in the response to question 4. For better understanding, for example, tightly pressed fingers against the skin will lead to a protruding part where bones are holding underneath. Even though the local inhomogeneity in stiffness for a PDMS sample is not as much as bones and skins, it is obvious enough since the difference of measured interfacial fluctuation is over 5 nm acquired from roughness data.
2. There may be some misunderstanding about causality. We were not intending to prove that solid tip highlights the stiffness difference of the sample rather than reflecting the real topography by imaging PDMS samples with possible inhomogeneous stiffness distribution at the nanoscale. The argument will be more evident if we replace ‘solid tip’ with ‘high stress level’, which make it as

“high stress level highlights the stiffness difference of the sample rather than reflecting the real topography”

This argument is logically correct since measured topography is directly related to the strain, which is inversely proportional to interfacial stiffness derivable from stress-strain relations in the definition of material modulus, i.e., $\sigma = M \cdot \varepsilon$ (σ , M and ε represent stress, stiffness and strain, respectively). It does not need further evidential justification from possible stiffness inhomogeneity but conversely serves in our manuscript as explicit evidence for the assumption of interfacial inhomogeneity. That is why we further stated in the manuscript that

“Since the measured area is identical for both tips and the stress is stable during the scanning process, the only accountable reason for the roughness difference is rooted in the material inhomogeneity at the nanoscale.”

3. Besides, from the formula discussed above, a minor difference in stiffness, i.e., M , will induce a significant change in strain ε when stress σ is stably maintained at a high level,

which means significant difference in imaging result does not correspondingly require the application of another copolymer possessing significant difference in stiffness distribution. On the other hand, even if we do so, the ideal experimental result will be an imaging difference in the material boundary. The results will not yield a particularly meaningful outcome other than the proof of high stress level posed by solid tips, which we have demonstrated for at least three times in the PDMS imaging and relevant calculations. Therefore, we do not think it is vital to perform additional experiments on such copolymer blocks.

Author's Response to Reviewer #3:

While the manuscript is improved in terms of clarity, the problem solved by the technological innovation is very minor and better suited for a more specialized journal such as Ultramicroscopy. I appreciate the shift of focus from tuning elastic parameters to energy absorption, but it is not clear from the data how important or useful of an advance this is. Further, even if this solves a problem for this mode of scanning probe microscopy, this is itself a highly specialized and uncommon variant of atomic force microscopy. While I do not recommend publication in Nature Communications, the authors should consider the following changes before moving forward elsewhere.

Response: Even though the reviewer did not recommend for the publication, we still would like to thank him so much for spending valuable time reviewing our work. The objections render us more objective on the analysis, evaluation and further promotion of the work.

1. In response to the questions on the innovation, we would like to briefly clarify the contribution and significance by quoting our comments in our previous cover letter:

“To the best of our knowledge, this is the first attempt that integrates cellular material into the SPM system for imaging optimization by exploiting its energy-absorbing capability established via programmable 3D-printing. This approach opens a new path towards the specific applications of cellular solids in the energy-absorption field and sheds light on novel SPM studies based on 3D-printed tips possessing exotic properties.”

2. The reviewer felt confused about the role and importance of energy-absorption characterization in this research. The energy-absorption characterization in the manuscript includes two main parts, i.e., instrumented indentation tests (IIT) and numerical analysis based on static and dynamic loading, which corroborates the mechanical behavior of CMA tips and instructs the geometrical design of the tips.

Specifically, the stress level (i.e., peak stress and plateau stress) and absorbed energy quantitatively obtained from engineering stress-strain response in IIT serve as the first approximation of tip design as we have stated in the manuscript:

“Assuming that in practical applications, the allowable stress level transmitted from the tip to the surface also increases as the sample itself becomes stiffer, stiffer samples requires the selection of a stiffer tip.”

Then numerical calculation based on dynamic impact further was presented, building a more accurate correspondence of specific stiffness relations between the sample and required CMA structure:

“This result translates into a specific requirement of stiffness-matching principle in the design of the CMA structures as a combined consideration of minimizing impact indentation based on the simulations while still keeping the tip deformation in a reasonable range needed to maintain its stability in specific scanning applications.”

Finally, finite element analysis based on static loading as well as IIT data was employed to calculate tip stiffness as we have stated in the manuscript:

“To predict the structure stiffness, static compressive analysis was performed with the apex subjected to uniaxial compression for specific displacements corresponding to applied stresses.”

In a word, the energy-absorption characterization reveals the mechanical behavior of tips during the impact and addresses why and how to design a proper CMA tip for better imaging.

3. The reviewer regarded this technology as a highly specialized and uncommon variant of AFM. We do not agree with this point. As far as we are concerned, the only difference between our experimental configuration and the ‘common’ scanning probe microscopic technologies in the reviewer's impression lies in our setup of a vertical tuning fork, which employs shear force as feedback. However, the position or the construction of a tuning fork itself should neither be the reason that makes it a highly specialized and uncommon SPM technique nor the disadvantages that reduce the innovative values or application potentials of CMA tips. The reason that we chose a vertical tuning fork for probe assembly was that it was currently the most convenient and fastest way to experimentally demonstrate the benefit from CMA tips. It does not mean the method cannot be extensively applied in other SPM fields, and it is not necessary to test the tips repeatedly on each single SPM device to verify the universality of their applications.

1) While I understand the title was changed to attempt to narrow the scope, it does the opposite. Scanning probe microscopy is a more general term than atomic force microscopy. I recommend making the title reflect the actual content of the paper, such as “3D printed cellular probes for gentle contact in shear-mode atomic force microscopy.”

Response: We would like to thank the reviewer for his kind advice. We believe that we have sufficiently explained the reason why we select 'SPM' in the title as well as the keywords in the response to the 2nd question raised by the second reviewer.

2) Shear force feedback is presented as a feature that is unique to tuning forks, which seems to ignore the great body of work studying lateral force microscopy using optical levers.

Response: We would like to clarify that there is no substantial evidence in our manuscript manifesting the application of CMA tips strictly requires the configuration of vertical tuning fork or shear force feedback. We have repeatedly emphasized that specific feedback mode, i.e., shear/normal force, optical lever, should not be the reason that restricts the application area of CMA tips. Conversely, CMA tips can be perfectly suitable in any SPM situations where mechanical contact or impact is inevitable, which is, in fact, independent of feedback modes. We have fully explained in the previous version of cover letter why vertical tuning fork based shear force feedback is more practically convenient than typical normal force feedback, which is quoted here:

“If normal-force feedback is applied, the attachment of fiber piece to the tuning fork would be difficult, especially a perfect normal direction is needed (Cover letter Fig.2c shown below). Besides, the location of tips under the microscope will be hard to confirm, which poses a challenge in practical scanning. Directly printing nanostructures to the tuning fork without using an optical fiber as media is not currently applicable because the chemicals used for resist developing would corrupt the components of the tip-mount. Since the normal impact (Cover letter Fig.2b, d) we were trying to optimize is independent of specific feedback modes, therefore we do not think it is necessary to perform the whole experiments again with normal force feedback.”

Cover letter Fig.2 Tip-mount configurations for shear force and normal force imaging mode. (a, b) Tuning fork photograph and operation diagram based on shear force imaging mode. (c, d) Tuning fork photograph and operation diagram based on normal force imaging mode.

Even though normal force feedback based CMA tip imaging is difficult to realize, it is still available. It is only a matter of fabrication techniques and operation skills, which cannot be interpreted as the incapability of the CMA tips. We are sorry that we cannot offer further exploration using optical levers for feedback limited by instrumental configurations, but we think it is unnecessary to laboriously test more tips in every single species or feedback mode of SPM devices in order to justify the generality of our creation.

3) At a high level, I understand the main virtue of the lattice being to buckle into a normal-force-limited state when the probe is put into contact for shear mode imaging. This leads to two major points (A) an image should be provided in the manuscript (Fig 1) that shows the geometry

during imaging and the expected configuration of the lattice during imaging and (B) this concept should be explored by imaging the same sample using multiple set points to evaluate how this buckling affects image quality.

Response:

1. We are grateful for this suggestion. For point A, we have considered this characterization in the early stage of the experiment, but we did not implement it for three reasons. First, the CMA structure was built on top of a fiber facet attached to a vertical tuning fork, which means the visible space of the tip will be confined within a length of $61\ \mu\text{m}$ (i.e., the height of CMA structure ($a=5\ \mu\text{m}$)) when the tip is approaching against the sample surface, as shown in Cover letter Fig. 3. It requires extremely laborious works on the assembly and calibration of optical components so that the microscopic details of the tip can be transmitted along a very harsh angle to the far-field in order not to be blocked by the optical fiber facet and finally captured by CCD camera (Cover letter Fig. 3). Even if this can be perfectly realized, the best image that could be expected will only provide particular information of local structure in a distorted way. Second, how to locate the tip position under optical objectives will be a big problem since the tip keeps approaching, retracting, and moving from point to point in milliseconds during scanning, which renders the adjusting of the optical path to acquire a clear and static picture technically impossible. Third, the scanning was maintained at a high rate of $6\sim 8\ \text{ms/point}$. This means the event of tip approaching or buckling will last at a duration no more than $1\sim 2\ \text{ms}$, which requires a specialized CCD camera currently unavailable to us. Even if all the listed problems could be entirely solved, in our opinion, the buckling images of a scanning tip cannot further provide valuable and informative value for instruction or reference other than the fact of tip buckling. Instead, the practical scanning experiments that we have conducted in various

substrate conditions and feedback parameters should be the center and focus that conclusively proves the whole creation more straightforwardly.

Cover letter Fig.3 Schematic of experimental configuration to obtain tip images during imaging.

2. For point B, we do agree with the reviewer. The imaging by solid tips using multiple set points has already been conducted, as shown in Supplementary Fig.9. The imaging results by CMA tips at multiple setpoint values are newly appended in Supplementary Fig.10.

4) The buckling of the lattice during contact likely leads to a decrease in the lateral resolution, which seems evident in Figures 3 and 4. It is not clear this tradeoff will be worth it for samples that are not damaged during imaging.

Response: For the imaging on silicon microgrid shown in Fig.3.a (ii) of the manuscript, the resolution and contrast have indeed decreased when selecting improper CMA tip ($a = 5 \mu\text{m}$) for scanning. However, the resolution and contrast were perfectly presented when using another CMA tip ($a = 1 \mu\text{m}$). In this test, the degradation of imaging quality using CMA tips was

attributed to mismatched stiffness that leads to an unstable scanning, especially crossing the steep slope patterns and, therefore, conversely proves the necessity of stiffness-matching rules for the geometrical design of CMA tips in order to avoid such an issue. It cannot be interpreted as evidence that all CMA tips will induce a decrease in resolution. On the contrary, the images scanned by properly designed CMA tip even showed a better quality compared with commercial tips, which can be demonstrated from manuscript Fig.3.a (i), Fig.4b, h, j and Supplementary Figure 4, 12. In a word, the “tradeoff” should not have existed with properly designed tips, and that is why we proposed stiffness-matching design rules and further employed numerical analysis as well as indentation tests to characterize and predict tip stiffness.

5) Given that the polymer tip itself is not very hard (speaking here of the material property hardness as opposed to using the term hard to mean stiff as is done in the manuscript), it is unlikely it will damage any samples besides polymers and biological materials. Given the low lateral resolution (in part due to the large tip diameter), it is not clear this imaging approach actually brings anything to the table that is not already possible using other modalities of scanning probe such as tapping mode imaging using standard optical lever based approaches.

Response: First, the creation of CMA tips is indeed mainly focused on imaging optimization on soft samples such as polymers and biological material by employing the energy-absorbing behavior during the scanning impact. In addition, the feasibility of scanning on stiffer samples such as silicon material was as well demonstrated in the manuscript. The only thing that limits sample selection is the achievable stiffness range of CMA structures, which could be readily settled by adjusting the shape and geometry of the microstructure itself. Second, we do not agree with the reviewer's comments on the low lateral resolution since the experimental results in our manuscript clearly suggest the opposite. Third, it is true that some specialized SPM technologies have employed advanced modalities that could mitigate the interaction force

between tip and sample. However, our work provided a general and alternative solution for more common technologies that are short of such advanced modalities by merely changing the construction of the tips. Similarly, a published paper titled “Multifunctional hydrogel nano-probes for atomic force microscopy” (Nat. Commun. 2016, 7, 11566) described the development of hydrogel tips for AFM. We believe the fact that machined silicon tips have long governed the commercial market and multiple functions can be realized using advanced modalities instead of hydrogel tips does not reduce the scientific contribution and significance of this research work. For the same reason, we insist on the novelty and contribution of our creation.

6) It is worth emphasizing for readers that this probe would not work in normal mode as the non-linear nature of the force-displacement curve would make it impossible to identify the sample height. This is a major limitation of this manuscript.

Response: We do not agree with the reviewer’s comments for three reasons. First, the compression stage of the CMA tips in the scanning is mostly distributed in the elastic regime with a linear mechanical response. Second, there is no direct causal link between the non-linear nature and inaccurate height measurement. Even if the tip works in a plastic deformation stage with a non-linear behavior, the compression at each point during the scan is identical since the stress level controlled by feedback parameters (i.e., setpoint, PGain and IGain) is stable, which suggests the difference of tip deformation would not significantly affect the measurement of sample morphology. As a clear proof contrary to the reviewer’s comments, we have experimentally demonstrated the scanning accuracy in the imaging of both silicon microgrid and PDMS patterns. The only problem for CMA tips working in a possible non-linear compression stage lies in the fact that the shape cannot completely recover to its original shape after each impact, which correspondingly degrades the energy-absorbing performance and thus

the optimization of scanning images. From our experimental experience, the CMA tips have a general lifespan of about a few weeks, which can be readily replaced thereafter using the convenient method introduced in the manuscript. Third, it is not logically reasonable to deduce the existence of such a normal mode restriction in the application of CMA tips according to our response to the 2nd question raised by the third reviewer.

Reviewer Comments

Reviewer #2 (Remarks to the Author):

The authors provided detailed rebuttal letter. Except one comment (Comment 4), the points raised in the previous round of review have been satisfactorily addressed. The quartz substrate of the PDMS sample is much harder than PDMS itself. If L2 is not much larger than (L1-L2), the stiffness of PDMS will be affected by the quartz substrate. The phenomenon that the measured value of the stiffness of the thin film sample is affected by the stiffness of the substrate has been confirmed by many researchers. If infinite flat blocks are added under the model given in Supplementary Figure 8 to support the stage pattern, the stiffnesses of the PDMS sample in the L1 and L2 parts will not be affected by the substrate, so the two are equal. This means that $\Delta I1 = \Delta I2$, and then according to Eq. (5): $\Delta h' = (L1 - \Delta I1) - (L2 - \Delta I2) = L1 - L2 = \Delta h$. In brief, the stiffness of the PDMS sample depends on the thickness of the sample. This conclusion can be easily verified by using CMA tips (especially solid tips) to measure the groove depth on PDMS samples with different thicknesses. Therefore, before this manuscript is published in NCOMMS, authors should discuss the specific thickness of PDMS samples since it affects the tip-sample interaction, interfacial deformation and scanning imaging. This discussion will also help readers better understand the experimental results.

Reviewer #3 (Remarks to the Author):

I am somewhat confused by the response to the last round of edits. My comments – and those of Reviewer 2 – push the authors into a more narrow scope to reflect the nature of their advance. Instead, they push back making claims that are not supported by experimental data.

Title:

Both myself and Reviewer 2 find SPM being too broad a term. The authors agree in their cover letter that this approach is not valid for all types of SPM or all types of AFM, so rather than writing a title that describes all possible applications of this approach, why not make the title reflective of what was actually studied in the paper? "Shear-mode tuning fork AFM" would be accurate as that is what is demonstrated in the paper.

Generality to other approaches:

In the prior round of revision, I contested that it would be challenging to use this approach in normal mode imaging. While the fact that normal mode imaging could in principle be done, the authors acknowledge that it is practically challenging and not demonstrated in the present manuscript. Further, a better distinction would be contact or intermittent contact imaging. The authors state in the cover letter:

"The only problem for CMA tips working in a possible non-linear compression stage lies in the fact that the shape cannot completely recover to its original shape after each impact, which correspondingly degrades the energy-absorbing performance and thus the optimization of scanning images."

This clearly restricts the present approach to contact mode imaging. This restriction should be clearly stated.

Resolution:

The authors state in the cover letter:

"Second, we do not agree with the reviewer's comments on the low lateral resolution since the experimental results in our manuscript clearly suggest the opposite."

While there is not a passage in the manuscript that states the resolution in clear terms, it seems that the smallest resolved features are ~40 nm, which is by no means high resolution in the world of AFM. High resolution probes in AFM are ~5 nm and can resolve features at the single

nanometer limit.

Depiction of Imaging:

I requested some depiction of the probe during imaging to show how the probe is used. While the authors replied with a length depiction of why this would be challenging to implement, a scheme would be acceptable. Critically, the value proposition of this approach is that the non-linear deformation of the lattice allows for gentle contact which can be leveraged for contact mode imaging. That central idea is not presented in any of the figures.

Author's Response to Reviewers General Note:

Dear reviewers,

We wish to thank you all for the time and effort that you have engaged in reviewing our paper. Your constructive comments have provided valuable insights for us to refine the contents and analysis substantially. In this document, we try to address the issues raised as best as possible with our itemized responses given directly afterward in red color. We sincerely look forward to meeting your expectations.

Once again, we are grateful for your precious input and we would like to acknowledge your contributions explicitly.

Thank you.

Best regards,

On behalf of all the co-authors

Author's Response to Reviewer #2:

The authors provided detailed rebuttal letter. Except one comment (Comment 4), the points raised in the previous round of review have been satisfactorily addressed. The quartz substrate of the PDMS sample is much harder than PDMS itself. If L2 is not much larger than (L1-L2), the stiffness of PDMS will be affected by the quartz substrate. The phenomenon that the measured value of the stiffness of the thin film sample is affected by the stiffness of the substrate has been confirmed by many researchers. If infinite flat blocks are added under the model given in Supplementary Figure 8 to support the stage pattern, the stiffnesses of the PDMS sample in the L1 and L2 parts will not be affected by the substrate, so the two are equal. This means that $\Delta I_1 = \Delta I_2$, and then according to Eq. (5): $\Delta h' = (L_1 - \Delta I_1) - (L_2 - \Delta I_2) = L_1 - L_2 = \Delta h$. In brief, the stiffness of the PDMS sample depends on the thickness of the sample. This conclusion can be easily verified by using CMA tips (especially solid tips) to measure the groove depth on PDMS samples with different thicknesses. Therefore, before this manuscript is published in NCOMMS, authors should discuss the specific thickness of PDMS samples since it affect the tip-sample interaction, interfacial deformation and scanning imaging. This discussion will also help readers better understand the experimental results.

Response:

We thank the reviewer for his insightful comments and suggestions. We have studied the comments carefully with our responses to this point listed below in detail.

1. About the thickness of PDMS samples

We agree with the reviewer's comment concerning the necessity of discussion of sample thickness. We have further supplemented the size information, photographic images and discussions in the manuscript method section and supplementary information. As shown in Supplementary Fig. 6b (appended below), the PDMS sample that we employed for the scanning tests is a bulk polymer solid with a size of $11.5 \times 10.5 \times 1.75 \text{ mm}^3$. Therefore, we have

$$L_1 - L_2 \approx 80 \text{ nm}$$

$$L_2 \approx 1.75 \text{ mm}$$

which suggests L_2 is almost 4 orders of magnitude larger than the stage height (i.e., $L_1 - L_2$).

Supplementary Fig. 6b Photographic images and size information of the PDMS sample used for scanning.

2. About the thin film effect in compression response

The stiffness of thin films can indeed be affected by the underlying substrate that possesses a different stiffness nature from the films as the reviewer commented, leading to a branch of stiffness-measurement researches on soft film/hard substrate or hard film/soft substrate (Ref: Composites Science and Technology, 2008, 68.1: 147-155). Since the ‘thin film’ is typically defined as a film with a thickness of microns and ‘very thin film’ typically has a thickness of hundreds of nanometers, it is apparently not the situation of our case given that $L_2 \approx 1.75 \text{ mm}$ as discussed above.

However, we would like to continue the discussion by presuming the scanned sample is a thin or very thin film. The film effect of soft film/ hard substrate in compression response is basically due to the severe constraints in the film posed by the relatively undeformable substrate during an indentation (Ref: Journal of Materials Research, 1999, 14.1: 292-301), in which circumstance, measured stiffness of film is actually the mechanical superposition of both film and its underlying substrate. However, the thin-film effect typically requires a certain

indentation depth, which is quantitatively represented by the ratio of indentation displacement to film thickness. We would like to directly cite some comments from relevant papers as follows about the relations between thin film effect and this ratio factor:

- a) “A commonly used rule of thumb suggests that substrate independent measurements can be obtained if the indentation depth is kept to less than **one-tenth** the film thickness.” (Ref: Journal of Materials Research, 1999, 14.1: 292-301)
- b) “When **$h_{max}/t = 0.05$** , the deviation of the calculated Young’s modulus, E , between the soft film and the soft bulk material, was almost zero. Moreover, the deviation was almost 5% when **$h_{max}/t = 0.10$** , and it reached 48% when $h_{max}/t = 0.45$.” (h_{max} denotes maximum indentation displacement and t represents film thickness) (Ref: Composites Science and Technology, 2008, 68.1: 147-155)
- c) “Specifically, this investigation also shows that for the situation of soft film/hard substrate combination, the indentation behavior is essentially identical if the modulus of the substrate is 10 times higher than that of the corresponding film and the response deviates consistently from that of bulk material with increasing of indentation depth. For penetration depth less than **10%** of the film thickness, the deviation could be acceptable.” (Ref: Thin Solid Films, 2008, 516.8: 1931-1940)
- d) “The standard rule is to limit the depth of indentation to one tenth of the film thickness to avoid the impact of the substrate response. ...For nanoindentation depths larger than about **1/10** of the thickness, the extracted Young's modulus is affected by the elastic response of the silicon substrate” (Ref: Thin Solid Films, 2009, 518.1: 260-264)

Therefore, a ratio factor of 10% is typically taken as a threshold value for acceptable thin film effect, below which, the indentation can be treated as a substrate-independent measurement. Based on our approximate estimation of tip stress as shown in the section ‘Estimation of stress level applied by CMA and solid tips’ in the supplementary discussion, the average stress level

posed by CMA tips is about 0.07 MPa, which induces a strain of PDMS sample of about $0.07 \text{ MPa} / 2.6 \text{ MPa} \approx 0.027$ (2.6 MPa is Young's modulus of PDMS material). Consequently, the ratio of indentation displacement to film thickness is about 2.7%, manifesting the scanning with CMA tips is substrate-free even if the sample is a thin or very thin film. For the solid probes used in PDMS imaging, the ratio factor is about 14.8%~27.6%. Therefore, the regardlessness of sample thickness can be treated as another advantageous benefit derived from the application of CMA tips.

Interestingly, the customized deformable CMA tips seem to provide an alternative solution to the thin film effect in indentation measurement. Typical indentation tests require a certain range of indentation displacement to ensure a stable and reasonable output of force measurement, which is restricted by the acceptable threshold of the indentation-film thickness ratio (i.e., 10%) especially when the film thickness is reduced to hundred nanometers, as Boé et al. has reported:

“It is commonly accepted that the results are reliable for films thicker than about 200 nm since the indentation depth must remain small compared to the film thickness while being large enough for not being below the resolution of the equipment and for avoiding surface artifacts” (Ref: Thin Solid Films, 2009, 518.1: 260-264)

This trade-off can be perfectly resolved by employing CMA tips as an indenter. The necessary displacement of indentation instrument needed for data collection will not solely be provided by sample deformation but mainly supported by tip deformation, in which case the indentation of the sample can be kept within a very small range avoiding the thin film effect while the whole displacement is large enough to generate convincing characterization. The film stiffness can be acquired by simply solving a mathematical model with measured force-displacement data and tailored stiffness of CMA tips.

3. About the deduction of measured height

The reviewer comments are cited below:

“If infinite flat blocks are added under the model given in Supplementary Figure 8 to support the stage pattern, the stiffnesses of the PDMS sample in the L1 and L2 parts will not be affected by the substrate, so the two are equal. This means that $\Delta l_1 = \Delta l_2$, and then according to Eq. (5): $\Delta h' = (L_1 - \Delta l_1) - (L_2 - \Delta l_2) = L_1 - L_2 = \Delta h$. In brief, the stiffness of the PDMS sample depends on the thickness of the sample.” (Supplementary Figure 8 has been numbered as Supplementary Figure 9 in the revised files)

Since the scanning result is independent of the underlying substrate in our case, the deformation model of the high and low stage can be simply treated as springs with different initial lengths under the same stress (Supplementary Fig. 9, appended below). The stiffness characterized by Young's modulus for both regions is indeed the same, which induces the same strain under the compression. Please note here the strain is the same while the displacement is not the same (i.e., $\Delta l_1 \neq \Delta l_2$) since the latter is defined as the product of initial length and the strain:

$$\Delta l_1 = L_1 * \varepsilon$$

$$\Delta l_2 = L_2 * \varepsilon$$

Δl_1 and Δl_2 denote the displacement of the high and low stage under compression, respectively. L_1 , L_2 corresponds to the initial length and ε represents the same strain. In a word, the difference of measured stage height is governed by the applied stress instead of sample thickness or assumed stiffness difference derived from the thin-film effect.

Supplementary Fig. 9 Morphological schematic of the high (blue) and low (red) stage of a step profile before and after the compression by assuming that material under the tip compression can be isolated from surrounding material and treated as springs in the calculation.

4. About the discussion of samples thickness

According to the discussions above, the imaging on the PDMS sample is not affected by its thickness since the sample is better considered as bulk material rather than film patterns and the tips work at a relatively low-stress level. We don't think it is necessary to continue the groove measurement with different sample thicknesses for a few reasons. First, it is technically impossible to strictly control thickness as the single variable in all the processes of mold fabrication and scanning. The template stripping method in the sample fabrication can not assure the two copies are exactly the same in micro features or micro composition distribution since it is affected by natural polymerization and manual stripping operation. Further, the scanning tips

should be calibrated with resonant frequency and phase feature before practical use, leading to slight approaching differences when scanning new samples if control parameters should be kept the same without any adjustment. Therefore, it is not rigorous to solely attribute any possible imaging difference to different sample thicknesses. Second, the assumed thickness problem is actually not a direct CMA tip related issue that uniquely arises as a result of CMA tip design and its implementation. It is better to be regarded as a part of the experimental method instead of the central topic of the discussion. However, we still supplemented all the expanded descriptions and discussions in the relevant parts (Manuscript Methods, Supplementary Fig. 6b and supplementary discussion entitled 'Estimation of stress level applied by CMA and solid tips') where we engaged in more details with sample thickness as the reviewer suggested for convincing proof and better illustration of the whole experimental results.

Author's Response to Reviewer #3:

I am somewhat confused by the response to the last round of edits. My comments – and those of Reviewer 2 – push the authors into a more narrow scope to reflect the nature of their advance. Instead, they push back making claims that are not supported by experimental data.

Title:

Both myself and Reviewer 2 find SPM being too broad a term. The authors agree in their cover letter that this approach is not valid for all types of SPM or all types of AFM, so rather than writing a title that describes all possible applications of this approach, why not make the title reflective of what was actually studied in the paper? “Shear-mode tuning fork AFM” would be accurate as that is what is demonstrated in the paper.

Response:

We thank the reviewer for this very valuable suggestion. After serious discussions with a few experts in this field from our laboratory, we are very sorry that the previous title, i.e., 3D-printed cellular tips for scanning probe microscopy, is indeed not accurate and rigorous enough to evaluate the whole research work. Therefore, in order to better define our study, including the central topic, the experimental method, especially the technology background, and to avoid any misunderstanding of the description by the readers, we sincerely accept the reviewer's proposal, changing the title to “3D-printed cellular tips for tuning fork atomic force microscopy in shear mode”. All relevant parts have been correspondingly revised and marked.

Generality to other approaches:

In the prior round of revision, I contested that it would be challenging to use this approach in normal mode imaging. While the fact that normal mode imaging could in principle be done, the

authors acknowledge that it is practically challenging and not demonstrated in the present manuscript. Further, a better distinction would be contact or intermittent contact imaging. The authors state in the cover letter:

“The only problem for CMA tips working in a possible non-linear compression stage lies in the fact that the shape cannot completely recover to its original shape after each impact, which correspondingly degrades the energy-absorbing performance and thus the optimization of scanning images.”

This clearly restricts the present approach to contact mode imaging. This restriction should be clearly stated.

Response:

We thank the reviewer for pointing this out.

1. About the normal mode imaging

The reason why we do not recommend to further perform normal-mode imaging is mostly due to the same normal contact shared with shear-mode imaging, which theoretically does not affect the energy-absorbing performance of CMA tips in practical imaging. Besides, it seems unnecessary to continue the exploration or discussion on normal mode imaging using CMA tips since we have narrowed the scope of content by directly indicating shear-mode imaging in the revised title. However, we would like to enrich the research system by implementing additional normal-mode imaging tests for convincing proof that corroborates the potential universality of tip applications. The results and discussions are appended in the supplementary discussion entitled ‘Normal-mode imaging with CMA tips’

Supplementary Fig. 16 Normal-mode imaging with CMA tips. **a, b** Photographic image of assembled normal-mode probe based on cantilevered tuning fork and the optical micrograph of its components, respectively. Scale bars are 2 mm for **a** and 200 μm for **b**. **c, d** Height images obtained from trace and retrace processes with solid tip (solid cone) and CMA tip ($a = 5 \mu\text{m}$), respectively. All scale bars are 3 μm .

As shown in Supplementary Fig. 16a, b, the optical fiber with tips at one facet was carefully attached to the cantilevered tuning fork with UV adhesive. The vibration of tuning fork renders intermittent normal contact between the tip and sample. The solid tip was employed for an imaging comparison with the CMA tip. The setpoint was controlled within a moderate range of 0.25~0.3. Supplementary Fig. 16c, d present representative height imaging details obtained from trace and retrace process with corresponding tips. The results from the solid tip clearly reveal a broadened groove width in contrast to the images acquired from the CMA tip, which coincides with the experimental results of shear-mode imaging revealed in the main manuscript.

Therefore, given the theoretical analysis as well as overall experimental results, the advantageous application of CMA tips independent of specific feedback mode can be confirmed.

2. About the restrictions

We acknowledge that the potential restrictions were not clearly stated in the manuscript. The imaging tests based on shear mode as well as normal mode belong to the intermittent contact. Theoretically, the imaging improvement using CMA tips relies on the microstructural energy-absorbing performance, which mitigates the interfacial impulsion exerted from tip to sample. It requires both deformation and recovery of struts in the scanning process. Therefore, it is technically suitable for this intermittent-contact based imaging that enables repetitive approach-retract processes for contact feedback at extremely high frequency. For typical direct contact imaging, the tips are kept pressing against the interface without recovery during the scanning process. Unlike the situation of intermittent contact, it theoretically has no dynamic approaching events in the scanning and therefore the impulsion applied to the sample interface is not significantly related to the microstructural construction of tips and the structure-derived energy-absorbing performance, in which case the implementation of CMA tip is no longer effectively applicable.

Typically, direct contact imaging requires the set-up of a reflective lever for feedback, which is not integrated into our scanning device system. We are sorry that we could not perform more tests on direct contact with CMA tips since the configuration of our scanning device is only restricted to intermittent contact imaging. Instead, as the reviewer suggested, we have clearly stated the restrictions of the tips in the supplementary discussion entitled 'Restrictions in the application of CMA tips'.

Resolution:

The authors state in the cover letter:

“Second, we do not agree with the reviewer's comments on the low lateral resolution since the experimental results in our manuscript clearly suggest the opposite.”

While there is not a passage in the manuscript that states the resolution in clear terms, it seems that the smallest resolved features are ~40 nm, which is by no means high resolution in the world of AFM. High resolution probes in AFM are ~5 nm or less and can resolve features at the single nanometer limit.

Response:

1. About relative resolution

We feel gratitude for the comment. The impression of high or low resolution is a relative concept. The divergence in the discussion of imaging resolution between dear reviewer and us basically comes from different standards for judgment. The reviewer employed commercial AFM probes for reference. The resolution of the commercial products can admittedly reach down to the atomic level, which is indeed relatively higher than the CMA tips. However, we selected solid tips and CMA tips with mismatched stiffness as a contrast to stiffness-matched CMA tips and the experimental results clearly suggest a higher resolution from the latter.

In our opinion, we feel more proper to evaluate the performance of CMA tips within our experimental system since the comparisons made in our experiments were based on the same condition, i.e., same imaging device with same feedback mode, same testing sample with same material composition, same scanning patterns and same direct laser writing employed for all tips manufacturing, while the proposed commercial AFM tips were performed in another device platform using different tips (e.g. different materials, aspect ratios), samples and measurement mechanisms.

Then the main topic of our study is not focused on further increasing the absolute resolution to a comparable or even better level than the commercial products. The absorbing-energy performance of CMA tips was utilized to mitigate the impulsions that could have distorted the surface and decreased the current resolution. Therefore, the absolute resolution limit from CMA tips is not primarily concerned and we have been making plenty of efforts demonstrating the microstructural effect of tips in precisely rebuilding a ‘tip-free’ interfacial morphology and maintaining the original resolution without further degradation instead of greatly increasing it, just as we have stated in the manuscript:

“Quantitatively, the sizes of the smallest distinguishable features are approximately 60~80 nm by the CMA tip, while the smallest features are distributed in the range of 300~600 nm using the solid tip. Considering that there are 500×500 points in a 21×21 μm² region, the step length is 42 nm. The spatial resolution of the CMA tip scanning result is rather close to the positioning limit based on the current configurations (see Supplementary Discussion of resolutions). Therefore, the precise measurements and significantly improved image contrast (Fig.4h, i) inaccessible by solely adjusting feedback control parameters (Supplementary Fig. 10) conversely demonstrate the effective contributions by the CMA designs.”

2. About absolute resolution

We have to admit the resolution of fabricated CMA tips is lower than the commercial tips. The measured apexes fabricated by DLW have an average radius of 70 nm with a minimum value of 47 nm, leading to an absolute resolution of approximately the same level, while the radius of commercial tips could reach 5~10 nm. This is a long-existing and inevitable defect inherently derived from the photolithography process of direct laser writing in tip fabrication, which, on one hand, is expected to be improved as the technology is developing, on the other hand, can be improved with post-manufacturing such as pyrolysis or reactive ion etching as

proposed in Supplementary Fig. 17, 19. The tip radius can be shrunk down to 17 nm after RIE. However, as we have discussed above, the resolution mostly governed by the tip size is not directly related to the design or implementation of microstructural tips where the contribution of our study have rooted. Therefore, in our opinion, it's probably inappropriate to separate the resolution individually as a key criterion to evaluate the research value. In order to help readers better comprehend the context, we have supplemented discussions about the resolution in the supplementary discussion entitled 'Restrictions in the application of CMA tips'.

Depiction of Imaging:

I requested some depiction of the probe during imaging to show how the probe is used. While the authors replied with a length depiction of why this would be challenging to implement, a scheme would be acceptable. Critically, the value proposition of this approach is that the non-linear deformation of the lattice allows for gentle contact which can be leveraged for contact mode imaging. That central idea is not presented in any of the figures.

Response:

We thank the reviewer for the kind suggestion. In fact, we have already schematically presented this central idea in manuscript Fig. 2e,f and Supplementary Movie 1 by simulating a dynamic mechanical response to an impact process. In manuscript Fig. 2e, the CMA tip is deformed in the impact leading to a decrease of overall stress as well as the indentation (manuscript Fig. 2f-i) compared to the solid tip (manuscript Fig. 2f-ii). The interfacial displacement and stress were calculated as shown in manuscript Fig. 2g, f, respectively. Nevertheless, we still supplemented an additional scheme shown in Supplementary Fig. 3 (appended below) as suggested to further clarify the idea.

Supplementary Fig. 3 Schematics of the mechanical response of tip and substrate to dynamic impact in the scanning process. a Scanning with the CMA tip. **b** Scanning with the solid tip (cone shape). All objects are displayed in a render style of wireframes.

Reviewer Comments:

Reviewer #2 (Remarks to the Author):

The authors consulted relevant literatures and gave a detailed answer to the Comment 4 in the second round of review. From the latest experiment data provided by the authors (sample thickness, the ratio of indentation depth to sample thickness), the core experiment of this manuscript (the experiment shown in Fig. 4) suffers from a serious flaw. The manuscript cannot be published on NCOMMS until this flaw is resolved.

As authors stated "... $L_2 \approx 1.75$ mm ... the ratio of indentation displacement to film thickness is about 2.7%, manifesting the scanning with CMA tips ... For the solid probes used in PDMS imaging, the ratio factor is about 14.8%-27.6% ...", it is easy to get that the indentation of the CMA tip I ($a=5$, $n=15$) is about 47.5 μm , and the indentation of the solid tip is greater than 259 μm .

(1) From Supplementary Fig. 18, we can estimate that the height of the solid cone tip before pyrolysis does not exceed 15 μm from the diameter of its bottom projection and the tip angle (a taper angle of 28°) given by the authors. However, the indentation of this tip is not less than 259 μm , which is more than ten times of the height of the solid tip. This means the PDMS sample is in contact with the end face of the fiber during the experiment, which is an important factor in that the image quality of the tip (Fig. 4c) is worse than that of the CMA tip I (Fig. 4b).

(2) The authors also have the same problem as in (1) when comparing the image quality of CMA tip I ($a=5$ μm , $n=15$, left figure in Fig. 4j) and CMA tip II ($a=2$ μm , $n=15$, right figure in Fig. 4j). The tip height of CMA tip II is about 25 μm and the indentation of this tip is greater than CMA tip I (47.5 μm). Therefore the PDMS sample is also in contact with the end face of the fiber during the experiment.

The authors should ensure that the depth of the indentation is less than the height of the tip in the experiment, so as to draw convincing conclusions for readers. At the same time, since the indentation depth of PDMS in the experiments are relatively large, the authors also needs to ensure that the shape of the tip is consistent. That is, the shape of the solid tip must be a regular tetrahedron instead of a cone tip.

Other comments:

Since the indentation instrument needs to know the exact indentation depth of the sample, the deformation of the CMA tip cannot help to obtain the indentation depth. Therefore, based on the existing experimental results, it is not sufficient to conclude that "This trade-off can be perfectly resolved by employing CMA tips as an indenter."

Reviewer #3 (Remarks to the Author):

The authors have addressed the comments raised in the last round of review and I have no further technical objections to publication.

Author's Response to Reviewer #2 :

The authors consulted relevant literatures and gave a detailed answer to the Comment 4 in the second round of review. From the latest experiment data provided by the authors (sample thickness, the ratio of indentation depth to sample thickness), the core experiment of this manuscript (the experiment shown in Fig. 4) suffers from a serious flaw. The manuscript cannot be published on NCOMMS until this flaw is resolved.

As authors stated "... $L_2 \approx 1.75 \text{ mm}$... the ratio of indentation displacement to film thickness is about 2.7%, manifesting the scanning with CMA tips ... For the solid probes used in PDMS imaging, the ratio factor is about 14.8%-27.6% ...", it is easy to get that the indentation of the CMA tip I ($a=5, n=15$) is about $47.5 \text{ }\mu\text{m}$, and the indentation of the solid tip is greater than $259 \text{ }\mu\text{m}$.

(1) From Supplementary Fig. 18, we can estimate that the height of the solid cone tip before pyrolysis does not exceed $15 \text{ }\mu\text{m}$ from the diameter of its bottom projection and the tip angle (a taper angle of 28°) given by the authors. However, the indentation of this tip is not less than $259 \text{ }\mu\text{m}$, which is more than ten times of the height of the solid tip. This means the PDMS sample is in contact with the end face of the fiber during the experiment, which is an important factor in that the image quality of the tip (Fig. 4c) is worse than that of the CMA tip I (Fig. 4b).

(2) The authors also have the same problem as in (1) when comparing the image quality of CMA tip I ($a=5 \text{ }\mu\text{m}, n=15$, left figure in Fig. 4j) and CMA tip II ($a=2 \text{ }\mu\text{m}, n=15$, right figure in Fig. 4j). The tip height of CMA tip II is about $25 \text{ }\mu\text{m}$ and the indentation of this tip is greater than CMA tip I ($47.5 \text{ }\mu\text{m}$). Therefore the PDMS sample is also in contact with the end face of the fiber during the experiment.

The authors should ensure that the depth of the indentation is less than the height of the tip in the experiment, so as to draw convincing conclusions for readers. At the same time, since the indentation depth of PDMS in the experiments are relatively large, the authors also needs to ensure that the shape of the tip is consistent. That is, the shape of the solid tip must be a regular tetrahedron instead of a cone tip.

Other comments:

Since the indentation instrument needs to know the exact indentation depth of the sample, the deformation of the CMA tip cannot help to obtain the indentation depth. Therefore, based on the existing experimental results, it is not sufficient to conclude that “This trade-off can be perfectly resolved by employing CMA tips as an indenter.”

Response: We thank the reviewer for the good comments and appreciate the rigorous attitudes. Since all the questions are essentially related to the high-stress level that leads to seemingly unreasonable strain, ratio factor (i.e., the ratio of indentation displacement to film thickness) as well as the derived displacement, we would like to address all the issues in one response as best as we could. Before that, we deeply apologize for the false statement about ‘ $L_2 \approx 1.75 \text{ mm}$ ’ in our last response since theoretically the effective compressible range, i.e., L_2 , in a dynamic impact is far less the whole sample thickness due to gradual propagation of stress waves. Therefore, it is inappropriate to apply the short-term dynamic stress or the transient ratio factor localized within a limited impact region for a solution of long-term global substrate (i.e., PDMS sample) response under constant (quasi-)static loading by employing overall sample thickness for calculation. For simplicity, the quasi-static loading is also classified as static loading in the context due to similar features of mechanical response. The issue is addressed in details from four aspects as follows:

Experimental evidence against excessive indentation depth:

The actual indentation displacement of sample cannot reach a depth of hundreds of microns, which can be proved by two pieces of experimental evidence:

1. The indentation of tips cannot theoretically reach a depth of 259 μm in the deceleration phase after approaching the interface since the maximum step range for the scanner in z -direction is 30 μm . Besides, the loading rate (i.e., the initial speed of impact) needed to achieve such a distance within the deceleration period is far beyond the mechanical capability of the piezoelectric scanner.
2. If the indentation displacement reaches hundreds of microns, the fiber end face instead of the tip will be in direct contact with the surface as indicated by the reviewer. The interfacial deformation of the soft sample will be large enough in three dimensions to be perceived by the unaided eye. However, we didn't observe such a visible macro deformation in all the scanning tests. Further, the reviewer thought the contact of the fiber end might be the reason for worse imaging quality. In fact, the imaging capability will be directly disabled under such circumstances instead of merely decreasing the imaging quality. The contact of the impact will be edges of the fiber facet or the whole facet to the PDMS sample surface. The shape of the contact area cannot provide a stable perception of an approaching state and cannot resolve the spatial features of the microgrooves at all since the circular fiber facet has a large radius of 62.5 μm . Even though the imaging quality was degraded as shown in manuscript Fig.4c, the microfeatures can still be identified, which clearly contradicts the above inference, suggesting the hypothesis of hundreds of microns' indentation is not the actual situation and therefore cannot account for the worse imaging quality presented in Fig.4c.

Based on the above-mentioned evidence, it's doubtful about the correctness of the calculated stress level of different tips that were used to deduce the ratio factors. However, the

stress level was calculated by filling the reasonable spring model with the actual imaging outcomes (The number of collected data was raised to 30 for accuracy in our last revision). The high-stress level is absolutely required to induce such a large strain at the interface that leads to a significant difference in height measurements by two tips. In our opinion, the deduction of the stress level is quite convincing according to relevant references (*J. Mater. Res.*, 2011, 26.6: 727).

The reason for seemingly unreasonable stress level——impact dynamics:

The calculation for global indentation displacement of the sample corresponding to the high-stress level or large ratio factor suggested by the reviewer was based on a prerequisite of static indentation, in which case the stress is presumed to constantly compress the sample with global interaction. The term ‘static’ means there is an instantaneous correspondence between the stress and its deformation, which has no spatial and temporal delay in the response and is not changing over space and time. However, the actual process of tip loading should be classified as a dynamic impact rather than a static loading, in which case the local transient physical quantity doesn’t have an absolute correspondence with the global response. The physical difference between dynamic and static loading is better described by referring the comments from a published book named ‘Introduction to Impact Dynamics’ (*Wiley*, 2017) as follows:

‘In the case of a structure under quasi-static loading, as the loading speed is very slow, any portion of the structure can be regarded as being in an equilibrium state that does not vary with time. However, in the case of a structure under dynamic loading, the loading speed is fast and/or the external load varies with time. It will take time for the disturbance caused by the external load to reach different parts of the structure. In general, therefore, the various portions of the structure will be in a non-equilibrium state.’

Three features can be concluded from relevant knowledge background:

#1: The deformation process under dynamic loading is much more complex than the static loading. The pressure of the stress wave and the indentation displacement are not uniformly distributed on the global scale and vary with distance (from the interaction position) and time. (*Int. J. Impact Eng.*, 2008, 35.9: 1063-1074)

#2 The impact-induced stress dominates a short-term transient substrate response localized within the near-field region of impact instead of a global one. (*Int. J. Impact Eng.*, 2017, 99: 111-121)

#3: The dynamic loading generally features an overpressure of the stress at the beginning of the contact and a decrease afterward. (*Mater. Sci. Eng., A*, 2011, 528.27: 7857-7866)

Point #1 and #2 can be found in a large number of references. Specifically, in our study, we have also presented these features in the simulation results based on dynamic loading as shown in manuscript Fig. 2f (shown below in Response Fig.1), which clearly illustrates the effective region of interaction and non-uniform local distribution of stress and displacement varying with interaction duration.

Response Fig.1 Temporal evolutions of interfacial stress and displacement distribution of the soft sample during the impact of a solid tip in z -direction (tip is concealed). Scale bars are 100 nm. Source: manuscript Fig.2f-ii.

The feature of Point #3 is also supported by many studies, which can be easily understood as a result of intensive momentum change at a transient period. It is also presented in our manuscript as we have stated:

‘...unreal grains rise due to occasional overstressed situations...’

Point #3 implies short-term local stress at a high level could indeed happen as in our case even though the magnitude may seem so unreasonable at the first sight that would have caused a significant global displacement response under static loading without considering stress distribution in space and actuation duration.

Full explanation of the displacement-related issues based on dynamic impact mechanism

The ‘ratio factor’ defined by the ratio of indentation displacement to film thickness numerically equals the local strain in response to the local stress. Therefore, the fundamental problem of all the concerns raised by dear reviewers essentially lies in how to correctly calculate the displacement based on the given strain. Clearly, the answer is different for static and dynamic loading. For the static loading, the original dimension of the strain has a global scale. However, for the latter, the above features of impact dynamics suggest the derivation of strain from interfacial distortions presented in our supplementary discussions is a temporal solution to the local response within the ‘near-field’ region of the interaction position where the compressive response first reaches an equilibrium state, manifesting only the compression in this near-field range can be regarded as a static response and the effective compressible dimension for local compression should be accordingly a measure of the near-field scale instead of the global size. In other words, the outcome of the stress levels, the strains and the ratio factors yielded from our model and imaging measurements are approximations averaged in the time of deceleration period and the space of the near-field region, which are not appropriate to

be used as physical quantities constantly effective in the overall space and time for the solution of a global response.

Response Fig.2 Schematic of the transient near-field response of PDMS pattern to dynamic impact. Source: Supplementary Fig. 9.

Specifically, the compressive spring was proposed as an ideal mechanical model to approximately estimate local stress by separating the near-field region for analysis, as shown in Response Fig.2, where major deformative responses took place. Therefore, even though the PDMS sample has a thickness of millimeters, the effective range of the dynamic compression (i.e., the proposed 'film thickness' in the case of dynamic impact) as shown in Response Fig.1 used to calculate the indentation displacement (by multiplying film thickness by the proposed ratio factor) is far less than whole sample thickness considering the transient deceleration period (i.e., the sample compressed by solid tip roughly has a compressible dimension no more than several microns derived from numerical simulation, instead of a macroscopic deformative scale

of sample thickness (i.e., 1.75 mm) as was adopted by the reviewer). That is, derived from a temporal local stress, the effective ‘film thickness’ of the ratio factor in the dynamic impact case is also presented as a short-term local quantity with limited range, which only enables the approximation of the local displacement but cannot be substituted with the overall sample thickness for global calculation. Here we apologize again for the incorrect statement about ‘ $L_2 \approx 1.75 \text{ mm}$ ’, the false marks presented in the schematic of the spring model in our last revision, and the misuse of ‘film thickness’ in dynamic impact process that may lead to your misunderstandings. These mistakes are carefully corrected in this revision.

Besides, unlike the typical dynamic impact tests with a constant loading rate, the tip is decelerated by tip cushioning as well as device control in the impact process of the scanning course, which means the overpressured stress wave is not likely to constantly continue spreading to the global leading to a macroscopic deformation (e.g. displacement of hundreds of microns), since, on one hand, the supplying mechanical energy in the system is limited as the tip is decelerated and stopped while the overpressured stress keeps fast attenuating (typically exponentially) with the propagating distance in the far-field region contributing to far less displacement than in the near-field, on the other hand, the deformative response in the far-field region will also decay rapidly due to material condensing as well as surface stretching.

We believe the above-discussions fully address why the effective compressible range, i.e., L_2 , should have a local scale instead of sample thickness. Furthermore, a brief overview of the overall dynamic impact process is concluded for your kind consideration as follows:

1. Overpressured stress is yielded within the interfacial region excessively compressing this near-field region at the beginning of the impact, which decreases with time and distance. As an average approximation in this interaction period as well as the effective compressible range, the stress level (the one calculated from imaging results in the

supplementary discussion) induces transient large strain in the near-field scale, causing a visual distortion of the interfacial patterns as shown in manuscript Fig.4c.

2. The impact-induced stress waves keep propagating to the far-field as the tip is decelerating and finally stopped in this period of milliseconds, causing compressions of the further area but fast decaying with the propagating distance.
3. As the tip finally stops, no energetic stress is further produced any longer to continuously support the entire compression on the global scale while the finite energy of on-going stress waves is soon dissipated in the mechanical interaction. The interfacial stretching recovers as the tip is subsequently retracted for another scanning course.

Brief responses and Defects in the last revision

Based on all the discussions above, for question #1 and #2 raised by the dear reviewer, the experimental evidence has been presented suggesting the impossibility of excessive indentation depth while the ‘abnormal’ stress, ratio factor and corresponding displacement are indeed reasonable by substituting the micron-scale of local compressible range for global sample thickness in relevant calculations according to the principle of impact dynamics. Besides, an indentation depth of several hundreds of microns is unlikely since the corresponding interfacial stretching will directly lead to such an excessive distortion of surface patterns that the nanofeatures cannot be resolved. Based on the calculated stress level, numerical analysis, approaching speed as well as impact duration, the maximum local indentation depth for solid tips will be hundreds of nanometers approximately depending on the actual approaching speed, which coincides with all the experimental results. The CMA tips exhibit a much smaller indentation depth due to its cushioning property, yielding more accurate and reliable images without excessive local mechanical interactions.

For the last comments by the reviewer, the application of CMA tips as an indenter is technically possible since the deformation of tips can be determined from the measured stress and the pre-calibrated tip stiffness and thus the indentation depth of the sample can be calculated. However, this issue is no longer important, since, strictly speaking, our discussions of static instrumented indentation by analogy with the dynamic scanning imaging process are not rigorous due to the difference in the mechanical mechanism, which are therefore deleted in this revision to avoid misunderstandings.

For the possible influences on imaging from PDMS sample thickness concerned by the reviewer in the last round of review, we have clarified no such influence due to the macroscopic size of the PDMS sample in the last response. Additionally, the above-discussed impact dynamics can serve as a supplementary explanation since the impact-induced transient local displacement of the sample is roughly four orders of magnitude less than the overall size. Besides, all the imaging experiments of the PDMS patterns were strictly performed on the same PDMS sample (Supplementary Fig. 6b) and substrate underneath, which further excludes the existence of sample thickness as a potential variable affecting the imaging quality. Some preliminary scanning tests with the 3D-printed tips at the early stage of the experiments also achieved correct imaging results even though based on different samples with different macroscopic thicknesses (not shown in the manuscript), which convinced us the independency of imaging results from sample thickness.

Finally, we acknowledge that there are some defects in the discussions of stress approximation and film-thickness-related indentation issues in our last revision. First, the interactions were not clearly stated from the perspective of dynamic impact, which leads to misunderstandings of the stress outcomes. Second, the extended discussions of static-based instrumented indentation tests with CMA tips in light of a local stress value derived from dynamic impact process is inappropriate, under which circumstances the concept of ratio factor

and film thickness have been misused since the film thickness is literally a global concept different from the near-field compressible range in the case of dynamic impact.

We apologize for our carelessness. We have removed the inappropriate discussions and correspondingly revised relevant contents. We ensure the validity of all experimental results that are carefully verified, calculated, and analyzed based on hundreds of scanning tests. We deeply appreciate your precious suggestions and sincerely look forward to your approval, and thank you so much for the time and effort that you have engaged in reviewing our paper.

Reviewer Comments:

Reviewer #2 (Remarks to the Author):

The authors addressed my last concerns about the indentation depth and revised the misleading statement in the manuscript. Therefore, I recommend publishing this manuscript in NCOMMS.

Author's Response to Reviewers #2:

The authors addressed my last concerns about the indentation depth and revised the misleading statement in the manuscript. Therefore, I recommend publishing this manuscript in NCOMMS.

Response: We thank the reviewer for the encouraging comments.